# Evolutionary rescue of phosphomannomutase deficiency in yeast models of human disease

Ryan C Vignogna[1†], Mariateresa Allocca[2,3], Maria Monticelli[2,3,4], Joy W Norris[5], Richard Steet[5], Ethan O Perlstein[6], Giuseppina Andreotti[2*], Gregory I Lang[1*]

[1]Department of Biological Sciences, Lehigh University, Bethlehem, United States; [2]Institute of Biomolecular Chemistry, National Research Council, Pozzuoli, Italy; [3]Department of Environmental, Biological and Pharmaceutical Sciences and Technologies, University of Campania "Luigi Vanvitelli", Caserta, Italy; [4]Department of Biology, University of Napoli "Federico II", Napoli, Italy; [5]JC Self Research Institute, Greenwood Genetic Center, Greenwood, United States; [6]Perlara PBC, Berkeley, United States

**\*For correspondence:**
gandreotti@icb.cnr.it (GA);
glang@lehigh.edu (GIL)

**Present address:** [†]Department of Molecular Biology and Genetics, Weill Institute for Cell and Molecular Biology, Cornell University, New York, United States

**Abstract** The most common cause of human congenital disorders of glycosylation (CDG) are mutations in the phosphomannomutase gene *PMM2,* which affect protein *N*-linked glycosylation. The yeast gene *SEC53* encodes a homolog of human *PMM2*. We evolved 384 populations of yeast harboring one of two human-disease-associated alleles, *sec53*-V238M and *sec53*-F126L, or wild-type *SEC53*. We find that after 1000 generations, most populations compensate for the slow-growth phenotype associated with the *sec53* human-disease-associated alleles. Through whole-genome sequencing we identify compensatory mutations, including known *SEC53* genetic interactors. We observe an enrichment of compensatory mutations in other genes whose human homologs are associated with Type 1 CDG, including *PGM1*, which encodes the minor isoform of phosphoglucomutase in yeast. By genetic reconstruction, we show that evolved *pgm1* mutations are dominant and allele-specific genetic interactors that restore both protein glycosylation and growth of yeast harboring the *sec53*-V238M allele. Finally, we characterize the enzymatic activity of purified Pgm1 mutant proteins. We find that reduction, but not elimination, of Pgm1 activity best compensates for the deleterious phenotypes associated with the *sec53*-V238M allele. Broadly, our results demonstrate the power of experimental evolution as a tool for identifying genes and pathways that compensate for human-disease-associated alleles.

## Editor's evaluation

This valuable paper shows that experimental evolution can shed new and unbiased light on mutations involved in human diseases by showing how growth defects can be compensated. The evidence is convincing, benefiting from not only genetics but also well-established biochemical assays. This paper will be of interest to a broad group of evolutionary biologists and biologists interested in human diseases.

## Introduction

Protein glycosylation is an important cotranslational and posttranslational modification involving the attachment of glycans to polypeptides. These glycans play vital roles in protein folding, stability, activity, and transport (*Varki, 2017*). Glycosylation is one of the most abundant protein modifications,

with evidence indicating that over 50% of human proteins are glycosylated (*Roth et al., 2012*; *Wong, 2005*). Despite this, we lack a global understanding of the complex pathologies involving protein glycosylation.

Congenital disorders of glycosylation (CDG) are a group of inherited metabolic disorders arising from defects in the protein glycosylation pathway, including *N*-linked glycosylation, *O*-linked glycosylation, and lipid/glycosylphosphatidylinositol anchor biosynthesis (*Chang et al., 2018*). *N*-linked CDG are further categorized into two groups based on the affected process: synthesis and transfer of glycans (Type 1) or processing of protein-bound glycans (Type 2) (*Aebi et al., 1999*). Mutations in the human phosphomannomutase 2 gene (*PMM2*) cause Type 1 CDG and are the most common cause of CDG (*Ferreira et al., 2018*). PMM2 forms a homodimer and catalyzes the interconversion of mannose-6-phosphate and mannose-1-phosphate (M1P) (EC 5.4.2.8). M1P is then converted to GDP-mannose, a required substrate for *N*-linked glycosylation glycosyltransferases.

Two of the most common pathogenic *PMM2* variants found in humans are p.Val231Met and p.Phe119Leu. The V231M protein is less stable and exhibits defects in protein folding (*Citro et al., 2018*; *Silvaggi et al., 2006*), whereas the F119L protein is stable but exhibits dimerization defects (*Andreotti et al., 2015*; *Kjaergaard et al., 1999*; *Pirard et al., 1999*). The budding yeast *Saccharomyces cerevisiae* contains a homologous phosphomannomutase enzyme encoded by the gene *SEC53*. *SEC53* is essential in yeast and expression of human *PMM2* rescues lethality (*Hansen et al., 1997*; *Lao et al., 2019*). Previously, *Lao et al., 2019* constructed strains harboring the yeast-equivalent mutations of the most common human-disease-associated *PMM2* alleles including V231M (*sec53*-V238M) and F119L (*sec53*-F126L).

Using experimental evolution, we sought to identify compensatory mutations able to overcome glycosylation deficiency. Previous studies have used experimental evolution of compromised organisms, to study evolutionary outcomes and identify compensatory mutations (*Harcombe et al., 2009*; *Helsen et al., 2020*; *Laan et al., 2015*; *Michel et al., 2017*; *Moser et al., 2017*; *Szamecz et al., 2014*). Experimental evolution allows for a more robust approach than traditional suppressor screens, enabling the identification of weak suppressors and compensatory interactions between multiple mutations (*Cooper, 2018*; *LaBar et al., 2020*).

Here, we present a 1000-generation evolution experiment of yeast models of PMM2-CDG. We evolved 96 populations of yeast with wild-type *SEC53*, 192 populations with the *sec53*-V238M allele, and 96 populations with the *sec53*-F126L allele for 1000 generations. We sequenced 188 evolved clones to identify mutations that arose specifically in the populations harboring the *sec53* human-disease-associated alleles. We find an overrepresentation of mutations in genes whose human homologs are other Type 1 CDG genes, including *PGM1* (the minor isoform of phosphoglucomutase), the most commonly mutated gene in our experiment. We show that evolved mutations in *PGM1* restore protein *N*-linked glycosylation and alleviate the slow-growth phenotype caused by the *sec53*-V238M disease-associated allele. We performed genetic and biochemical characterization of the *pgm1* mutations and show that these mutations are dominant and allele-specific *SEC53* genetic interactors.

## Results

### Experimental evolution improves growth of yeast models of PMM2-CDG

Mutations in the phosphomannomutase 2 gene, *PMM2*, are the most common cause of CDG. *Lao et al., 2019* constructed yeast models of PMM2-CDG by making the yeast-equivalent mutations for five human-disease-associated alleles in *SEC53*, the yeast ortholog of *PMM2*. These strains have growth rate defects that correlate with enzymatic activity and promoter strength. Increasing expression levels of the mutant *sec53* alleles twofold, by replacing the endogenous promoter (p*SEC53*) with the *ACT1* promoter (p*ACT1*), modestly improves the growth rate of *sec53* disease-associated alleles (*Figure 1A*).

We chose to focus on *sec53*-V238M and *sec53*-F126L because they are two of the most common disease alleles in humans and because the mutations have distinct and well-characterized effects on protein structure and function (*Figure 1A*; *Andreotti et al., 2015*; *Briso-Montiano et al., 2022*; *Citro et al., 2018*; *Kjaergaard et al., 1999*; *Pirard et al., 1999*; *Silvaggi et al., 2006*). We evolved 96 haploid populations of each mutant *sec53* genotype (p*SEC53*-*sec53*-V238M, p*ACT1*-*sec53*-V238M,

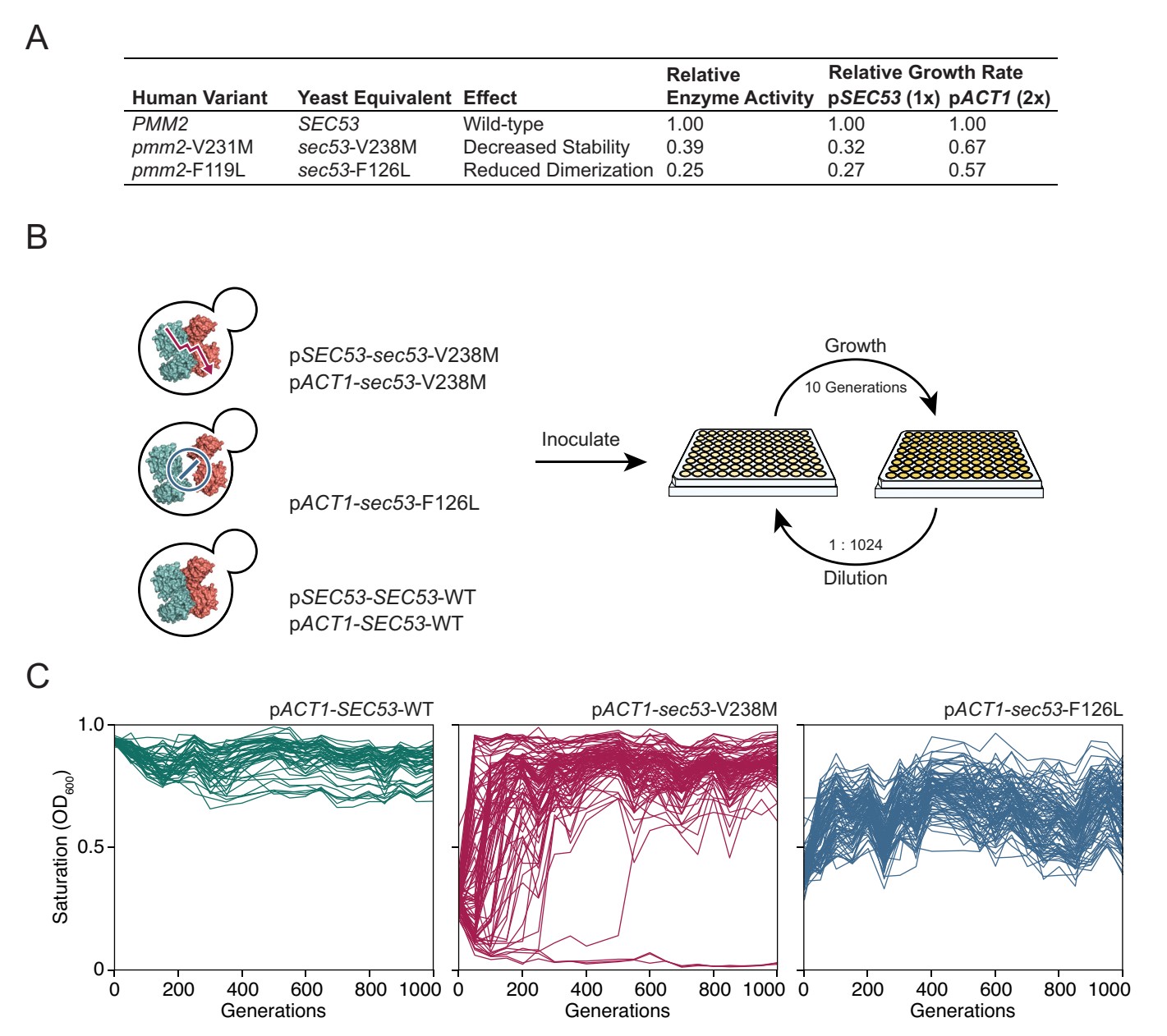

**Figure 1.** Experimental evolution of yeast models of congenital disorders of glycosylation. (**A**) Table of the various *sec53* alleles used in this study. In vitro enzymatic activities of Pmm2 are relative to wild-type Pmm2 and were previously reported (***Pirard et al., 1999***). Relative growth rates of yeast carrying various *SEC53* alleles were previously reported (***Lao et al., 2019***). (**B**) Diagram of the evolution experiment. Yeast carrying *sec53* mutations implicated in human disease were used to initiate replicate populations in 96-well plates: 96 populations of each mutant genotype and 48 populations of each wild-type genotype. Populations were propagated in rich glucose media, unshaken, for 1000 generations. (**C**) Single time-point $OD_{600}$ readings of populations were taken every 50 generations during the evolution experiment as a measure of growth rate. Each line represents one population.

The online version of this article includes the following figure supplement(s) for figure 1:

**Figure supplement 1.** $OD_{600}$ readings of each population.

and pACT1-sec53-F126L) and 48 haploid populations of each wild-type *SEC53* genotype (p*SEC53*-*SEC53*-WT and p*ACT1*-*SEC53*-WT) for 1000 generations in rich glucose medium (***Figure 1B***). Strains with the p*SEC53*-*sec53*-F126L allele grew too slowly to keep up with a $1:2^{10}$ dilution every 24 hr. Every 50 generations we measured culture density (single time-point $OD_{600}$) for each population as a proxy for growth rate (***Figure 1C***; ***Figure 1—figure supplement 1***).

Populations with the p*ACT1-sec53*-V238M allele show a range of dynamics, with some populations reaching maximum saturation ($OD_{600} \approx 1.0$) early in the experiment, while some do not even by Generation 1000 (*Figure 1C*). Three p*ACT1-sec53*-V238M populations went extinct before Generation 200. While saturation levels of p*ACT1-sec53*-F126L populations also increased over the course of the evolution experiment, none reached an $OD_{600}$ close to 1.0, indicating that these populations are likely less-fit than the evolved p*ACT1-sec53*-V238M populations. Each p*SEC53-sec53*-V238M population started the evolution experiment with growth rates comparable to *SEC53*-WT populations, indicating compensatory mutation(s) likely arose in the starting inocula (*Figure 1—figure supplement 1*). We therefore exclude these populations from subsequent analyses. Together, our data show that the PMM2-CDG yeast models acquired compensatory mutations throughout the evolution experiment and that the extent of compensation depends on the specific disease-associated allele.

## Putative compensatory mutations are enriched for other Type 1 CDG-associated homologs

To identify the compensatory mutations in our evolved populations, we sequenced single clones from 188 populations isolated at Generation 1000 to an average sequencing depth of ~50×. These included 91 p*ACT1-sec53*-V238M clones, 36 p*ACT1-sec53*-F126L clones, 32 p*SEC53-SEC53*-WT clones, 11 p*ACT1-SEC53*-WT clones, and 18 of the initially suppressed p*SEC53-sec53*-V238M clones (*Supplementary file 1*). Autodiploids are a common occurrence in our experimental system (*Fisher et al., 2018*; *Johnson et al., 2021*). We attempted to avoid sequencing autodiploids by screening our evolved populations for sensitivity to benomyl, an antifungal agent that inhibits growth of *S. cerevisiae* diploids more severely than haploids (*Venkataram et al., 2016*).

We find that the average number of de novo mutations (single nucleotide polymorphisms [SNPs] and small indels) varied between *SEC53* genotypes, with the p*ACT1-sec53*-F126L clones accruing more mutations per clone (8.14 ± 1.00, 95% confidence interval [CI]) compared to p*ACT1-sec53*-V238M (5.76 ± 0.54) and *SEC53*-WT clones (combined p*SEC53-SEC53*-WT and p*ACT1-SEC53*-WT) (6.49 ± 0.90) (*Figure 2—figure supplement 1*). Despite screening for benomyl sensitivity, most of the sequenced clones (131/188) appear to be autodiploids, with evidence of multiple heterozygous loci. In addition, we detect several copy number variants (CNVs) and aneuploidies shared between independent populations (*Figure 2—source data 1*, *Figure 2—source data 2*).

We identified putative adaptive targets of selection as genes with more mutations than expected by chance across replicate clones. Some of these common targets of selection are not specific to *sec53*. For example, mutations in negative regulators of Ras (*IRA1* and *IRA2*) are found in clones from each experimental group and are also observed in other laboratory evolution experiments across a wide range of conditions (*Figure 2A*; *Fisher et al., 2018*; *Gresham and Hong, 2014*; *Johnson et al., 2021*; *Kvitek and Sherlock, 2013*; *Lang et al., 2013*; *Venkataram et al., 2016*).

We identified putative *sec53*-compensatory targets as genes mutated exclusively in p*ACT1-sec53*-V238M and/or p*ACT1-sec53*-F126L populations (*Figure 2A*). The most frequently mutated genes among p*ACT1-sec53*-F126L clones are common targets in other evolution experiments such as *IRA1*, *IRA2*, and *KRE6*, suggesting that mutations capable of compensating for Sec53-F126L dimerization defects are rare or not easily accessible. In contrast, while recurrently mutated genes in p*ACT1-sec53*-V238M populations include common targets, we also identified a number of unique targets of selection, most notably in homologs of other Type 1 CDG-associated genes. We find an enrichment of mutations in CDG homologs in p*ACT1-sec53*-V238M clones (*Figure 2B*; *Figure 2—source data 3*).

Across all sequenced populations we identified ~620 nonsynonymous mutations. As there are nearly 6000 yeast genes, the probability of any given gene receiving even a single nonsynonymous mutation is low. We are therefore well powered to detect enrichment of mutations, but underpowered to detect depletion of mutations. To gain statistical power we aggregated our *SEC53*-WT data with three other large datasets using similar strains and propagation regimes (*Fisher et al., 2018*; *Johnson et al., 2021*; *Lang et al., 2013*). In this aggregate dataset, we observe a statistically significant depletion of mutations in Type 1 CDG homologs, suggesting that they are typically under purifying selection in experimental evolution (*Figure 2B*).

One of the CDG homologs, *PGM1*, is the most frequently mutated gene among p*ACT1-sec53*-V238M clones. *PGM1* encodes the minor isoform of phosphoglucomutase in yeast (*Bevan and Douglas, 1969*). We only find *PGM1* mutations in p*ACT1-sec53*-V238M populations which suggests

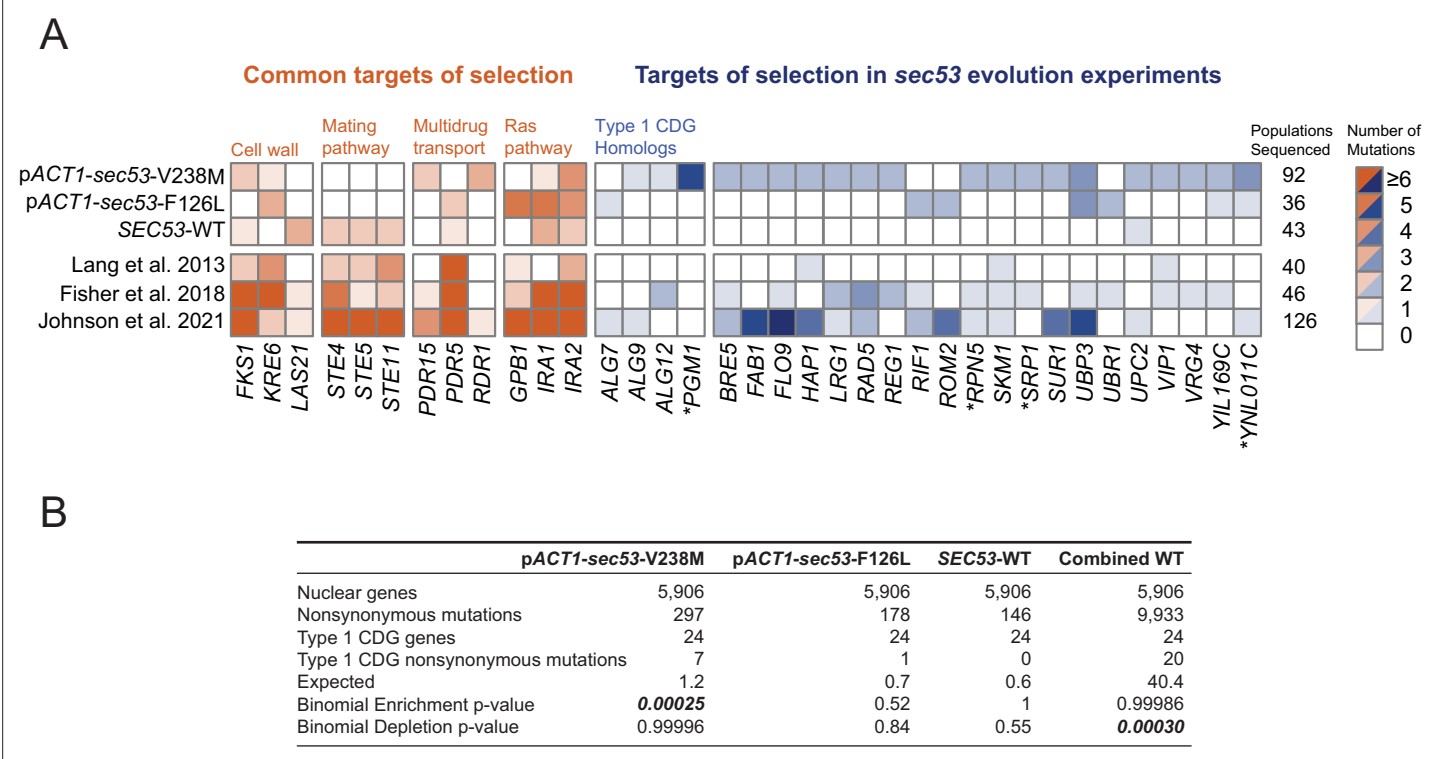

**Figure 2.** Mutations in Type 1 congenital disorders of glycosylation (CDG) homologs are enriched in *sec53*-V238M populations. (**A**) Heatmap showing the number of nonsynonymous mutations per gene that arose in the evolution experiment. Genes with two or more unique nonsynonymous mutations (and each Type 1 CDG homolog with at least one mutation) are shown. p*SEC53-SEC53*-WT and p*ACT1-SEC53*-WT are grouped as '*SEC53*-WT'. For comparison, we show data from previously reported evolution experiments where the experimental conditions were identical to the conditions used here, aside from strain background and experiment duration (bottom three rows). Commonly mutated pathways are grouped for clarity. Asterisks (*) indicate previously known *SEC53* genetic interactors (*Costanzo et al., 2016*; *Kuzmin et al., 2018*). (**B**) Binomial test for enrichment or depletion of mutations in Type 1 CDG homologs based on the number of nonsynonymous mutations observed in each experiment, the total number of yeast genes (5906), and the number of genes that are Type 1 CDG homologs (24). 'Combined WT' includes the *SEC53*-WT populations as well as three additional datasets: *Lang et al., 2013*, *Fisher et al., 2018*, and *Johnson et al., 2021*.

The online version of this article includes the following source data and figure supplement(s) for figure 2:

**Source data 1.** Table of recurrent copy number variants (CNVs) and aneuploidies in the evolution experiment.

**Source data 2.** Coverage plots of evolved clones.

**Source data 3.** Table of mutations in Type 1 congenital disorders of glycosylation (CDG) homologs.

**Figure supplement 1.** The number of de novo mutations per *SEC53* genotype.

their compensatory effects are unique to the *sec53*-V238M mutation and not glycosylation deficiency in general. Each of the five mutations in *PGM1* are missense mutations, rather than frameshift or nonsense. This is consistent with selection acting on alteration-of-function rather than loss-of-function (LOF; posterior probability of non-LOF = 0.96, see Methods).

To identify the compensatory mutation(s) that arose prior to the start of the evolution experiment in the initially suppressed p*SEC53-sec53*-V238M populations, we analyzed the 18 sequenced clones. For each clone, we find both wild-type *SEC53* and *sec53*-V238M alleles among sequencing reads. This could result from integration or maintenance of the covering plasmid. No sequencing reads align to the *URA3* locus of our reference genome, suggesting that the plasmid marker was lost, as expected following counterselection.

## Compensatory mutations restore growth

In order to validate putative compensatory mutations we reconstructed evolved mutations in *PGM1* and *ALG9*, two genes whose human homologs are implicated in Type 1 CDG (*Frank et al., 2004*; *Timal et al., 2012*). These heterozygous evolved mutations were constructed in a homozygous

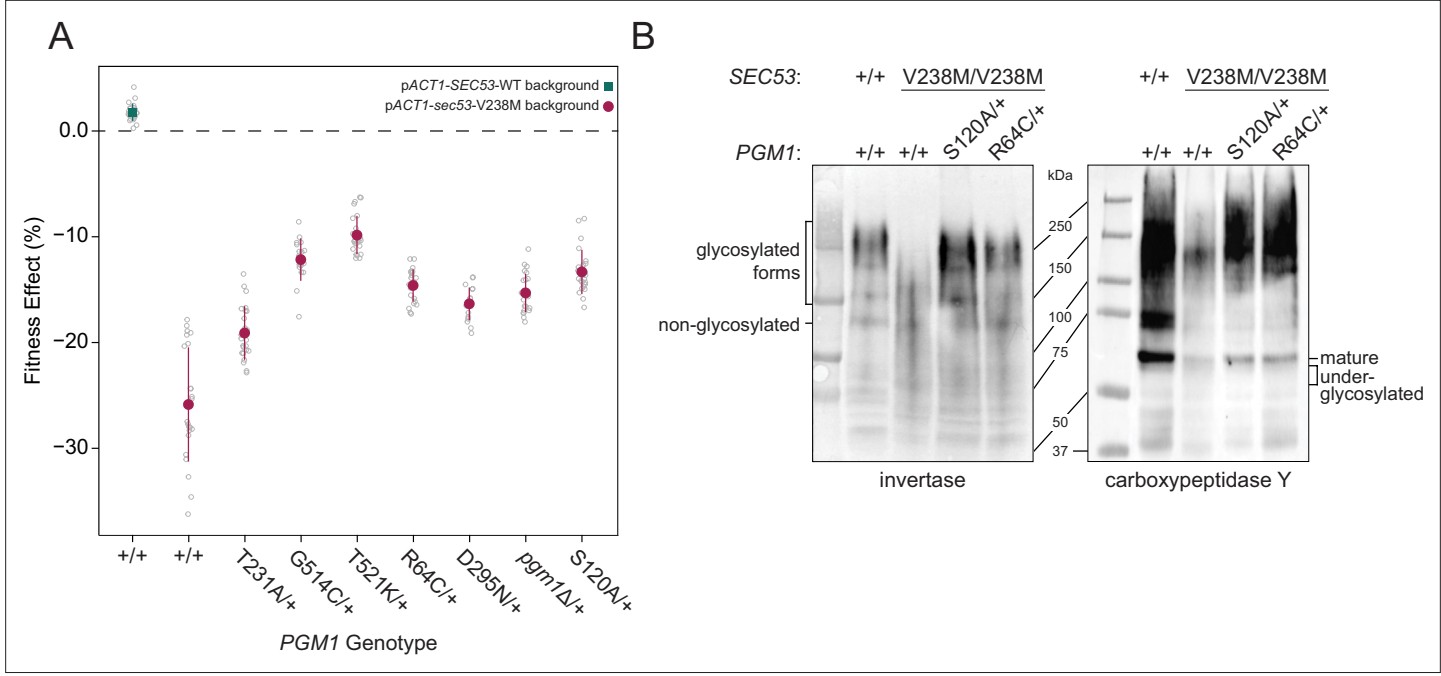

**Figure 3.** Evolved mutations rescue fitness and protein glycosylation defects of p*ACT1-sec53*-V238M. (**A**) Average fitness effects and standard deviations of reconstructed heterozygous *pgm1* mutations. Fitness effects were determined by competitive fitness assays against a fluorescently labeled version of the diploid p*ACT1-SEC53*-WT ancestor. Replicate measurements are plotted as gray circles. Pairs of *pgm1* fitness effects with non-statistically significant differences: R64C-D295N, R64C-*pgm1*Δ, R64C-S120A, D295N-*pgm1*Δ, G514C-T521K, G514C-S120A, and *pgm1*Δ-S120A (df = 194, F = 208.1, each p > 0.05, one-way analysis of variance [ANOVA] with Tukey post hoc test). (**B**) Western blots of invertase (left) and carboxypeptidase Y (right) from ancestral and reconstructed strains. In panels A and B, plus signs (+) indicate wild-type alleles. Genotypes are either homozygous wild-type (+/+), homozygous mutant (mutation/mutation), or heterozygous (mutation/+).

The online version of this article includes the following source data and figure supplement(s) for figure 3:

**Source data 1.** Raw images of blots and Ponceau stains.

**Figure supplement 1.** Fitness effects of reconstructed mutations and evolved clones containing those same mutations.

**Figure supplement 2.** Fitness effects of *pgm1* mutations in the *SEC53*-WT background.

**Figure supplement 3.** Fitness effects of homozygous *pgm1* mutations.

**Figure supplement 4.** Effects of *pgm1* mutations on invertase glycosylation in the p*ACT1-sec53*-V238M background.

**Figure supplement 4—source data 1.** Raw images of blots and Ponceau stains.

**Figure supplement 5.** Effects of *pgm1* mutations on invertase and carboxypeptidase Y (CPY) glycosylation in the p*ACT1-SEC53*-WT background.

**Figure supplement 5—source data 1.** Raw images of blots and Ponceau stains.

**Figure supplement 6.** Effects of *pgm1* mutations on invertase glycosylation in the p*ACT1-sec53*-F126L background.

**Figure supplement 6—source data 1.** Raw images of blots and Ponceau stains.

p*ACT1-sec53*-V238M diploid background (note that although each *pgm1* and *alg9* mutation assayed here arose in haploid-founded populations, they arose as heterozygous mutations in autodiploids). We quantified the fitness effect of each mutation using a flow cytometry-based fitness assay, competing the reconstructed strains against a fluorescently labeled, diploid version of the p*ACT1-SEC53*-WT ancestor.

The p*ACT1-sec53*-V238M ancestor has a fitness deficit of −25.86 ± 2.3% (95% CI) relative to wild-type. We find that each of the five evolved *pgm1* mutations are compensatory in the p*ACT1-sec53*-V238M background. The fitness effects of the *sec53/pgm1* double mutants range between −19.10 ± 1.07% and −9.84 ± 0.77% (i.e., the *pgm1* mutations confer a fitness benefit between 6.76% and 16.02% in the p*ACT1-sec53*-V238M background) (*Figure 3A*). We also find that the evolved *alg9*-S230R mutation is compensatory, as the *sec53/alg9* double mutant has a fitness effect of −17.89 ± 0.59% (*Figure 3—figure supplement 1*). We compared the fitness of these reconstructed double mutants to the fitness of the evolved clones which carry three to nine additional SNPs. In each case,

the evolved clones were more fit than the reconstructed strain, except for the evolved clone containing the *pgm1*-D295N mutation (*Figure 3—figure supplement 1*). Thus, while the *alg9* and *pgm1* mutations compensate for the *sec53*-V238M defect, other mutations contribute to the overall fitness of these evolved clones.

We constructed and assayed the fitness effects of the *pgm1* mutations in diploid p*ACT1-sec53*-F126L and p*ACT1-SEC53*-WT backgrounds. Each p*ACT1-sec53*-F126L strain was too quickly outcompeted by the reference strain to measure fitness, suggesting that the compensatory effects of the evolved *pgm1* mutations are specific for the *sec53*-V238M mutation or any fitness improvements in the p*ACT1-sec53*-F126L background are too minimal to detect. Each *pgm1* mutation is nearly neutral in the p*ACT1-SEC53*-WT background (*Figure 3—figure supplement 2*).

To determine if LOF of *PGM1* would phenocopy evolved mutations, we deleted one copy of *PGM1* (*pgm1Δ*) in the diploid p*ACT1-sec53*-V238M background and again measured fitness. We find that the heterozygous *pgm1Δ* mutation improves fitness (−15.28 ± 0.76%), indicating that LOF of *PGM1* is compensatory in the conditions of the evolution experiment (*Figure 3A*). It is not immediately clear, then, why we only identify missense mutations in *PGM1*. Each of the mutated Pgm1 residues is located around the active site of the enzyme, based on predicted protein structure (*Jumper et al., 2021*; *Stiers and Beamer, 2018*). It could be that the evolved *pgm1* mutations are LOF given this clustering but maintaining protein expression provides an ancillary benefit. To account for this possibility, we constructed a catalytically dead allele of *PGM1* by mutating the enzyme's catalytic serine (*pgm1*-S120A) (*Stiers et al., 2017a*). We find that *pgm1*-S120A improves p*ACT1-sec53*-V238M fitness (−13.32 ± 0.88%) and this does not significantly differ from *pgm1Δ* (*Figure 3A*), indicating that *PGM1* LOF is compensatory regardless of if that arises from loss of coding sequence or enzyme activity. We also measured fitness of homozygous *pgm1* mutations (*Figure 3—figure supplement 3*). In all cases, the fitness effect of heterozygous and homozygous *pgm1* mutants is not substantially different from each other, indicating that, by this assay, the evolved *pgm1* mutations are dominant in the p*ACT1-sec53*-V238M background, as are the *pgm1Δ* and *pgm1*-S120A mutations.

## Compensatory mutations restore protein glycosylation

We have established that *pgm1* mutations compensate for the growth rate defect of the *sec53*-V238M allele. To determine whether the evolved *pgm1* mutants also compensate for the molecular defects in *N*-linked glycosylation we examined two representative yeast glycoproteins, invertase and carboxypeptidase Y (CPY), in our reconstructed *pgm1* strains. Invertase forms both a non-glycosylated homodimer (120 kDa) and a secreted homodimer that is heavily glycosylated (approximate range of 140–270 kDa) (*Gascón et al., 1968*; *Zeng and Biemann, 1999*). Mature CPY only exists in its glycosylated form, with a molecular weight of 61 kDa (*Hasilik and Tanner, 1978*).

The ancestral p*ACT1-sec53*-V238M strain shows underglycosylation of invertase and CPY. We find a reduced abundance of the higher-molecular-weight glycosylated form of invertase and mature CPY, accompanied by the appearance of underglycosylated forms of these proteins (*Figure 3B*). We find that evolved *pgm1* mutations, *pgm1Δ*, and *pgm1*-S120A each restore glycosylation of these proteins to near-wild-type levels (*Figure 3B*, *Figure 3—figure supplements 4 and 5*). We do not observe rescue of protein glycosylation in p*ACT1-sec53*-F126L strains carrying *pgm1* mutations (*Figure 3—figure supplement 6*). Together, these suggest that *pgm1*-mediated rescue of p*ACT1-sec53*-V238M fitness is due to restoration of protein glycosylation and the effect is specific for the p*ACT1-sec53*-V238M background.

## Compensatory *PGM1* mutations are dominant suppressors of *sec53*-V238M

To further demonstrate that *pgm1*-mediated compensation is specific for *sec53*-V238M, we performed a genetic analysis by dissecting tetrads from strains that are heterozygous for both *sec53* and *pgm1*. In the absence of a compensatory mutation, we expect 50% large colonies and all tetrads showing a 2:2 segregation of colony size due to the single segregating locus (*SEC53/sec53*). However, if an evolved mutation is compensatory then we expect 75% large colonies with three types of segregation patterns: 2:2, 3:1, and 4:0, large to small colonies, respectively. These three segregation patterns are expected to follow a 1:4:1 ratio assuming no genetic linkage (*Figure 4A*). For each genetic test,

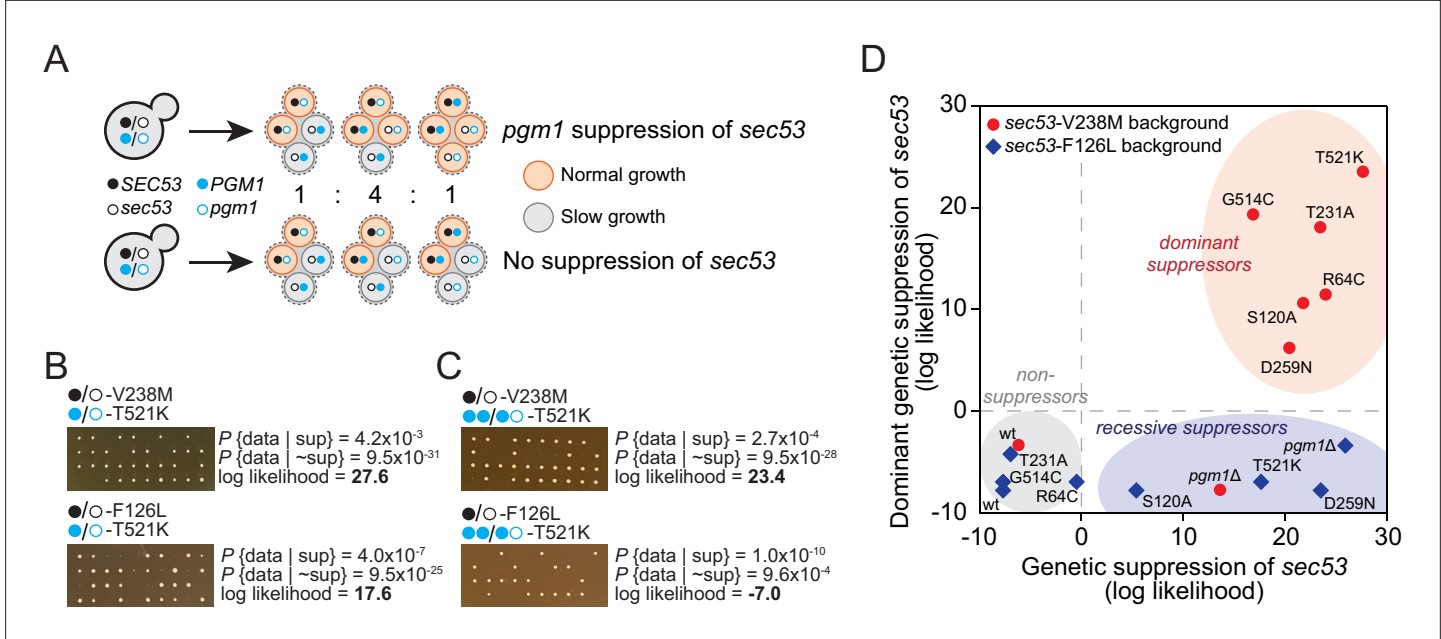

**Figure 4.** *pgm1* mutations are dominant suppressors of *sec53*-V238M. (**A**) Diagram of possible spore genotypes in the tetrad dissections. Heterozygous (e.g., *SEC53/sec53 PGM1/pgm1*) diploid strains could produce one of three tetrad genotypes based on allele segregation. (**B**) Example of *pgm1*-T521K dissections in a *sec53*-V238M background (top) and a *sec53*-F126L background (bottom). Listed are the probability and log likelihood of suppression, based on the ratio of normal growth (large colonies) to slow growth (no colonies or small colonies). (**C**) Example of *PGM1::pgm1*-T521K dissections in a *sec53*-V238M background (top) and a *sec53*-F126L background (bottom). (**D**) Plot of recessive suppression versus dominant suppression of the *pgm1* mutations in the *sec53*-V238M background (red circles) and *sec53*-F126L background (blue diamonds), based on the tetrad dissections.

The online version of this article includes the following source data and figure supplement(s) for figure 4:

**Source data 1.** Log-likelihood analysis of tetrad dissections.

**Figure supplement 1.** Tetrad dissections of *PGM1/pgm1* and *ALG9/alg9* strains.

**Figure supplement 2.** Tetrad dissections of *PGM1*-linked strains.

we dissected ten tetrads and performed a log-likelihood test to determine whether the segregation pattern is more consistent with genetic suppression than non-suppression (**Figure 4—source data 1**).

We find each of the evolved *pgm1* mutations compensates for the V238M allele of *SEC53* (**Figure 4—figure supplement 1**). For example, dissections of a strain heterozygous for p*ACT1-sec53*-V238M and *pgm1*-T521K show 78% large colonies (one 4:0 and nine 3:1, with a log-likelihood ratio of 27.6, **Figure 4B**). These genetic tests corroborate the results from the fitness assays by showing that all five evolved *pgm1* mutations, the *pgm1Δ* and *pgm1*-S120A mutations, as well as the evolved *alg9* mutation all suppress *sec53*-V238M (**Figure 4—figure supplement 1**). We next assayed each of the *pgm1* mutations in a *sec53*-F126L background. In contrast to the fitness assays, we find several *pgm1* alleles (T521K, D295N, S120A, and *pgm1Δ*) that suppress *sec53*-F126L (**Figure 4—figure supplement 1**). It is worth noting, however, that the degree of compensation is less than in the *sec53*-V238M background based on colony sizes.

To test if *pgm1* mutations are dominant, we integrated a second wild-type copy of *PGM1* to each of the strains such that each haploid spore will contain either two wild-type copies of *PGM1* (*PGM1::PGM1*) or both wild-type *PGM1* and mutant *pgm1* (*PGM1::pgm1*). We find that the five evolved *pgm1* mutations and *pgm1*-S120A, but not *pgm1Δ*, are dominant suppressors of *sec53*-V238M in the genetic assay (**Figure 4C, D**; **Figure 4—figure supplement 2**). For the *sec53*-F126L background, we find no dominant suppressors.

## Reduction of Pgm1 activity alleviates deleterious effect of PMM2-CDG in yeast

To determine how the evolved mutations alter Pgm1 enzymatic activity, we cloned mutant and wild-type *PGM1* alleles into bacterial expression vectors, purified recombinant enzyme, and assayed

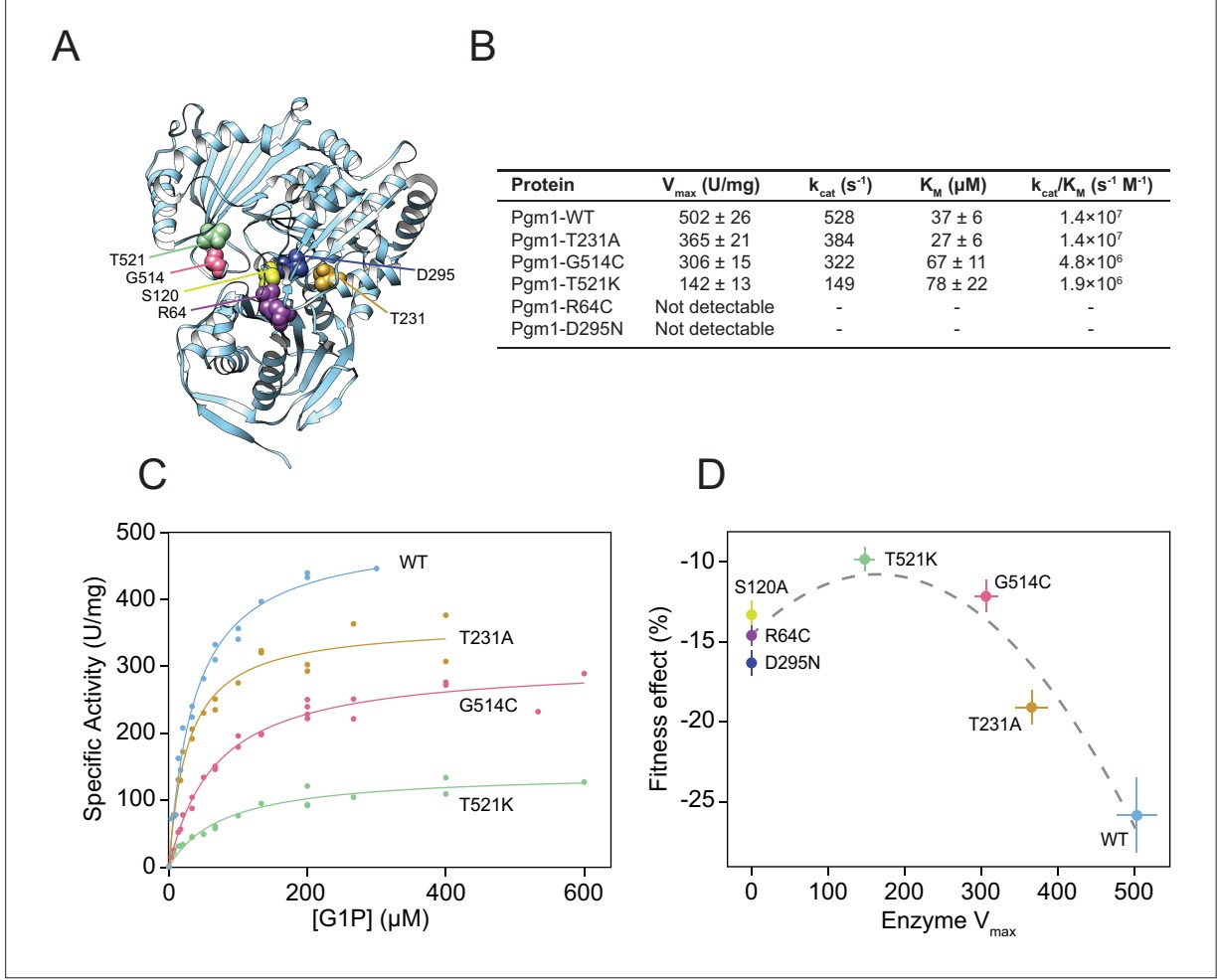

**Figure 5.** Complete loss of Pgm1 activity overshoots a fitness optimum. (**A**) AlphaFold structure of *S. cerevisiae* Pgm1 (*Jumper et al., 2021*). Mutated residues shown as spheres. (**B**) Kinetic parameters of recombinant Pgm1 enzymes as determined by a coupled enzymatic assay. Values ± 95% confidence intervals. (**C**) Michaelis–Menten curves of wild-type and mutant Pgm1. Replicate measurements are plotted as circles. (**D**) Pgm1 enzyme $V_{max}$ versus the corresponding allele's fitness effect in the diploid p*ACT1-sec53*-V238M background (as shown in *Figure 3A*). Horizontal and vertical error bars represent 95% confidence intervals. Best fit regression shown as a dashed curve ($y = 14.583 + 0.04613x − 0.00014x^2$, $R^2 = 0.907$, df = 4, F = 37.34, p = 0.0258, analysis of variance [ANOVA]). Note that we did not measure Pgm1-S120A activity and assume null activity.

The online version of this article includes the following source data and figure supplement(s) for figure 5:

**Source data 1.** Other biochemical properties of mutant Pgm1.

**Figure supplement 1.** Enzymatic assays.

**Figure supplement 2.** Thermostabilities of mutant Pgm1.

**Figure supplement 3.** Phosphomannomutase activity of mutant Pgm1.

**Figure supplement 4.** *pgm1* mutations increase glucose-1,6-bisphosphate (G16P) levels in the p*ACT1-sec53*-V238M background.

**Figure supplement 5.** Representative forward/reverse phosphoglucomutase assay.

phosphoglucomutase activity using a standard coupled enzymatic assay (*Figure 5A*; *Figure 5— figure supplement 1*). We find that mutant Pgm1 have variable activities, ranging from near-wild-type (Pgm1-T231A) to non-detectable (Pgm1-R64C and Pgm1-D295N) (*Figure 5B, C*). The near complete loss of activity of Pgm1-D295N is unsurprising since D295 coordinates a $Mg^{2+}$ ion that plays an essential role during catalysis (*Stiers et al., 2016*) and we verify that all the active forms show very low activity in the presence of ethylenediaminetetraacetic acid (EDTA) (*Figure 5—source data 1*). Thermostability did not differ between enzymes with detectable activity (*Figure 5—figure supplement 2*).

Comparison of the kinetic parameters of active mutant enzymes shows a reduction of maximal velocity ($V_{max}$) between 27% and 72% relative to wild-type Pgm1. Based on the apparent Michaelis constant ($K_m$), substrate affinities of Pgm1-G514C and Pgm1-T521K are lower than wild-type Pgm1 ($t$ = 4.30 and 3.14, df = 35.78 and 20.47, p = 0.0001 and 0.0051, respectively, Welch's modified $t$-test). Pgm1-T231A shows an increased affinity compared to wild-type Pgm1, but this difference is not statistically significant ($t$ = 1.94, df = 34.68, p = 0.0601, Welch's). Together these indicate that the five evolved *pgm1* mutations have varying effects on enzymatic activity. The relationship between *pgm1* fitness and Pgm1 $V_{max}$ is best fit by a quadratic, rather than a linear, function (*Figure 5D*). Therefore, optimal fitness is attained by tuning down, but not eliminating, Pgm1 activity.

## Mutant Pgm1 does not have increased phosphomannomutase activity

Phosphoglucomutases can exhibit low levels of phosphomannomutase activity, and we reasoned that active-site mutations in Pgm1 could alter epimer specificity from glucose to mannose, directly compensating for the loss of Sec53 activity (*Lowry and Passonneau, 1969*). We tested this in two ways. First, we directly measured phosphomannomutase activity of wild-type Pgm1 and two mutant Pgm1, using a $^{31}$P-NMR spectroscopy-based assay (*Figure 5—figure supplements 1 and 3A*). We find that wild-type Pgm1 and Pgm1-G514C show comparably low phosphomannomutase activity, accounting for 3.5% and 3.0% of their phosphoglucomutase activity, respectively. We could not detect any phosphomannomutase activity for Pgm1-D295N. We also tested for phosphomannomutase activity indirectly by determining whether M1P acted as a competitive inhibitor in the phosphoglucomutase assay. If mutant Pgm1 had altered epimer specificity, we would expect M1P to compete with glucose-1-phosphate (G1P) for residency within the active site, leading to an apparent decrease in phosphoglucomutase activity. However, we find no effect of the addition of M1P on phosphoglucomutase activity for any of the Pgm1 enzymes tested (*Figure 5—figure supplement 3B*). Together, these results indicate that mutant Pgm1 does not have enhanced phosphomannomutase activity.

## Pgm1 mutations increase intracellular glucose-1,6-bisphosphate levels

Glucose-1,6-bisphosphate (G16P) is required as an activator of both Pgm1 and Pmm2 (Sec53) and can stabilize pathogenic variants of Pmm2 (*Monticelli et al., 2019*). The only known source of G16P in yeast is low-level dissociation during the phosphoglucomutase reaction of Pgm1 and Pgm2. We reasoned that evolved *pgm1* mutations may increase intracellular levels of G16P which, in turn, would stabilize Sec53-V238M. Each active form of Pgm1 exhibits barely detectable activity in the absence of G16P and we determined an apparent half maximal effective concentration ($EC_{50}$) of G16P for several Pgm1 enzymes (*Figure 5—source data 1*). Given that Pgm1 is the minor isoform of phosphoglucomutase in yeast, we also reasoned that it could act as a glucose-1,6-bisphosphatase, similar to human *PMM1* (*Veiga-da-Cunha et al., 2008*). We performed a specific glucose-1,6-bisphosphatase assay on wild-type and mutant Pgm1 enzyme (*Figure 5—figure supplement 1*). However, none of the mutant Pgm1 enzymes have detectable glucose-1,6-bisphosphatase activity.

We next assessed G16P levels for two *pgm1* mutant alleles in both the p*ACT1-sec53*-V238M and the p*ACT1-SEC53*-WT backgrounds (*Figure 5—figure supplement 4*). We find that strains heterozygous for *pgm1*-T521K or *pgm1*-D295N show statistically significant increases in the amount of G16P present compared to wild-type *PGM1*, in the *sec53*-V238M background ($W$ = 24 and 8, p = 0.003 and 0.0002, respectively, Mann–Whitney $U$-test). However, this difference is not significant in the *SEC53*-WT background ($W$ = 41 and 15, p = 0.35 and 0.078, respectively, Mann–Whitney).

Finally, we also reasoned that *pgm1* mutations could differently affect the forward and reverse directions of the phosphoglucomutase reaction, with possible effects on the flux of metabolites within the cell. We analyzed the forward and reverse reactions of Pgm1, Pgm1-T231A, and Pgm1-G514C using G1P or glucose-6-phophaste (G6P) as the substrate (*Figure 5—figure supplement 1*). We again used $^{31}$P-NMR spectroscopy, as G1P and G6P have clearly distinguishable signals (*Figure 5—figure supplement 5*). The ratio of forward to reverse reactions is 3.0 ± 1.2 and 4.3 ± 1.1 (± standard deviation) for Pgm1-T231A and Pgm1-G514C, respectively. Although a slightly higher ratio was obtained for wild-type Pgm1 (7.4 ± 3.1), the differences between wild-type and Pgm1-T231A or Pgm1-G514C are not statistically significant ($W$ = 1 and 2, p = 0.057 and 0.53, respectively, Mann–Whitney).

# Discussion

The budding yeast *S. cerevisiae* is a powerful system for studying conserved aspects of eukaryotic cell biology due to its fast doubling time, small genome size, genetic tractability, and the wealth of genomics tools available (*Botstein and Fink, 2011*). Many human genes are functionally identical to—and can complement—their yeast homologs (*Kachroo et al., 2015*). 'Humanizing' yeast by replacing a yeast ortholog with the human gene or making an analogous human mutation in a yeast ortholog enables functional studies of human disease (*Franssens et al., 2013*; *Laurent et al., 2016*). For example, Mayfield et al. characterized the severity of 84 alleles of human *CBS* (cystathionine-β-synthase) that are associated with homocystinuria, identifying those alleles where cofactor supplementation can restore high levels of enzymatic function (*Mayfield et al., 2012*). Gammie et al. characterized 54 *MSH2* variants associated with hereditary nonpolyposis colorectal cancer showing that about half of the variants have enzymatic function but are targeted for degradation (*Gammie et al., 2007*). Subsequent work showed that treatment with proteasome inhibitors restores mismatch repair and chemosensitivity in yeast harboring these disease-associated variants (*Arlow et al., 2013*). In additional to functional characterization, yeast models of human disease enable screens for genetic interactions, thereby uncovering new avenues for therapeutic intervention. Wiskott–Aldrich syndrome, for example, is due to mutations in *WAS*, the human homolog of *LAS17*. Using a temperature-sensitive lethal allele of *las17*, Filteau et al. isolated compensatory mutations in two yeast strains. The authors identified both common and background-specific mechanisms of compensation, including LOF of *CNB1* (Calcineurin B), which is phenocopied by inhibition with cyclosporin A (*Filteau et al., 2015*).

Here, we use experimental evolution to identify mechanisms of genetic compensation in humanized yeast models of PMM2-CDG. We evolved 96 replicate populations for 1000 generation in rich glucose medium using strains carrying wild-type *SEC53* (the yeast homolog of *PMM2*) and two human-disease-associated alleles (*sec53*-V238M and *sec53*-F126L). We show that compensatory mutations arose throughout the evolution experiment. Fitness gains were greater for populations with the p*ACT1*-*sec53*-V238M allele compared to the p*ACT1*-*sec53*-F126L allele, which showed modest gains in fitness (*Figure 1C*). This indicates that the number of compensatory mutations and/or the ameliorative effects of compensatory mutations are greater for the p*ACT1*-*sec53*-V238M allele compared to the p*ACT1*-*sec53*-F126L allele. Our sequencing and reconstruction experiments show that both explanations are true. While the p*ACT1*-*sec53*-V238M populations showed several unique targets of selection, the p*ACT1*-*sec53*-F126L populations mostly acquired mutations in targets of selection observed in other evolution experiments (*Figure 2A*). In addition, the genetic reconstruction experiments show that some of the *pgm1* mutations are recessive suppressors of *sec53*-F126L, whereas all five *pgm1* mutations are dominant suppressors of *sec53*-V238M (*Figure 4*). The latter is unsurprising given that each *pgm1* mutation arose as a heterozygous mutation in a *sec53*-V238M autodiploid background.

Twenty-four genes have been associated with Type 1 CDG in humans (*Aebi et al., 1999*). All but two Type 1 CDG genes have at least one functional homolog in yeast, and most map one-to-one (*Figure 2—source data 3*). We find an overrepresentation of mutations in Type 1 CDG homologs in the evolved p*ACT1*-*sec53*-V238M populations (*Figure 2B*). This enrichment is striking considering that we observe a significant depletion of mutations in these same genes aggregated across evolution experiments with wild-type *SEC53*, which suggests that they are normally under purifying selection. Specifically, we find five mutations in *PGM1*, two in *ALG9*, and one in *ALG12*. In addition, one *ALG7* mutation was identified in a single p*ACT1*-*sec53*-F126L population. Of these four, only *PGM1* is a known *SEC53* genetic interactor, and none are known Sec53 physical interactors. *ALG7* encodes an acetylglucosaminephosphotransferase responsible for the first step of lipid-linked oligosaccharide synthesis, downstream of *SEC53* in the *N*-linked glycosylation pathway (*Barnes et al., 1984*; *Kukuruzinska and Robbins, 1987*). *ALG9* and *ALG12* act downstream of *SEC53* and encode mannosyltransferases, responsible for transferring mannose residues from dolichol-phosphate-mannose to lipid-linked oligosaccharides (*Burda et al., 1999*; *Burda et al., 1996*; *Cipollo and Trimble, 2002*; *Frank and Aebi, 2005*). This oligosaccharide is eventually transferred to a polypeptide. Each of the evolved mutations in these genes is predicted to have deleterious effects on protein function based on PROVEAN scores (*Supplementary file 1*; *Choi and Chan, 2015*). This suggests that prodding steps in *N*-linked glycosylation alleviates the deleterious effect of *sec53* disease-associated alleles, perhaps by altering pathway flux to match lowered M1P availability. Evidence from human studies

supports the idea of downstream genetic modifiers affecting PMM2-CDG phenotypes. For example, there is not a direct correlation between clinical phenotypes and biochemical properties of PMM2 (*Citro et al., 2018*; *Freeze and Westphal, 2001*). One such modifier is a variant of the glucosyltransferase *ALG6*, which has been implicated in worsening clinical outcomes of PMM2-CDG individuals (*Bortot et al., 2013*; *Westphal et al., 2002*).

In addition to Type 1 CDG homologs, we identify several other classes of putative genetic suppressors. Both the *sec53*-V238M and *sec53*-F126L mutations retain partial functionality and *Lao et al., 2019* showed that increasing expression of these alleles improves growth. We, therefore, expected to find CNVs and aneuploidies as compensatory mutations. Though we do identify several strains that show evidence of CNVs at the *HO* locus on Chromosome IV, where the *sec53* alleles reside in our strains, karyotype changes are not a major route of adaptation (*Figure 2—source data 1*). Reduced function of mutant Sec53 could be due to destabilization and degradation of the protein. We identify recurrent mutations in the 26S proteasome lid subunit *RPN5* in the p*ACT1-sec53*-V238M background, in the E3 ubiquitin ligase *UBR1* in the p*ACT1-sec53*-F126L background, and in the ubiquitin-specific protease *UBP3* in both p*ACT1-sec53* backgrounds (*Figure 2*). In addition, we identify a mutation in the ubiquitin-specific protease *UBP15* in a single p*ACT1-sec53*-V238M clone. And finally, we observe multiple independent mutations in the known *SEC53*-interactors *RPN5*, *SRP1*, and *YNL011C* in *sec53* populations, which could have positive genetic interactions with *sec53* (*Figure 2*).

We find a high probability that the mutational spectrum of *PGM1* (five missense mutations, 0 frameshift/nonsense) resembles selection for non-LOF (posterior probability = 0.96). However, *PGM1* LOF (*pgm1Δ*) provides a fitness benefit indistinguishable from two evolved mutations (*pgm1*-R64C and *pgm1*-D295N) and higher than one evolved mutation (*pgm1*-T231A). Likewise, the *pgm1Δ* allele restores invertase and CPY glycosylation and growth in the tetrad analyses, albeit recessively. It is not immediately clear, therefore, why we do not observe frameshift/nonsense mutations in *PGM1* in the evolution experiment. We have previously shown that genetic interactions between evolved mutations are not always recapitulated by gene deletion (*Vignogna et al., 2021*). Given that most of the evolved clones containing *pgm1* mutations are more fit than the reconstructed strains, it is possible that other evolved mutations interact epistatically only with non-LOF *pgm1* mutations. However, there are no shared mutational targets (genes or pathways) among the evolved clones with *pgm1* mutations and only one evolved *pgm1* clone contains a mutation in a known *PGM1* interactor (*FRA1*). These suggest any putative genetic interactions with our evolved *pgm1* mutations are allele specific.

Using tetrad dissections, we show that each evolved *pgm1* mutation and the catalytically dead *pgm1*-S120A mutation are dominant suppressors of *sec53*-V238M. *pgm1Δ* is not a dominant suppressor based on the tetrad analyses, contrary to the fitness assays where the heterozygous *pgm1Δ* allele confers a fitness benefit comparable to several evolved mutations. This discrepancy likely results from differences in ploidy or growth medium of the two experiments—diploids in liquid culture versus haploids on agar plates. We also find that several *pgm1* mutations (T521K, D295N, *pgm1Δ*, and S120A) are recessive suppressors of the *sec53*-F126L allele in the tetrad dissections, whereas we were unable to detect any improvement using fitness assays. Three of these four mutations have no detectable enzymatic activity. However, the R64C allele, which also does not have detectable enzymatic activity does not suppress.

To test specific hypotheses for the mechanism of suppression of *sec53* disease alleles by mutation of *pgm1*, we characterized the biochemical properties of purified recombinant Pgm1 mutant proteins. We find that phosphoglucomutase activity ranges from non-detectable activity (R64C and D295N) to near wild-type (T231). Based on well-characterized human and rabbit PGM1 structures, R64C and D295N mutations likely affect active-site integrity (*Liu et al., 1997*; *Stiers et al., 2016*). R64 helps hold the catalytic serine (S120) in place and D295 coordinates a catalytically essential $Mg^{2+}$ ion. The T231A mutation, which occurs in a residue near the metal-binding loop, shows the highest activity levels among our mutant Pgm1 enzymes. The other two mutations with low but detectable activity, G514C and T521K, both occur in the phosphate-binding domain of Pgm1, a region important for substrate binding (*Beamer, 2015*; *Stiers et al., 2017a*; *Stiers et al., 2017b*; *Stiers and Beamer, 2018*). Comparing *pgm1* fitness effects versus their corresponding enzyme $V_{max}$ we find an inverted U-shaped curve, where reducing phosphoglucomutase activity improves fitness, but complete loss of activity overshoots the optimum (*Figure 5D*). Plotting fitness versus $k_{cat}/K_m$ results in a qualitatively

similar trend, although comparing $k_{cat}/K_m$ between mutant enzymes may not be appropriate (*Eisenthal et al., 2007*).

We tested several hypotheses involving Pgm1 alteration-of-function. First, the evolved Pgm1 proteins could directly complement the loss of Sec53 phosphomannomutase activity by switching epimer specificity from glucose to mannose. However, we do not observe increased phosphomannomutase activity in the Pgm1 mutant proteins, nor is this hypothesis consistent with our genetic data: direct complementation would be expected to suppress both the *sec53*-V238M and *sec53*-F126L alleles.

A second possible mechanism of suppression is increasing the production of G16P. G16P is an endogenous activator and stabilizer of Pmm2 (Sec53). Compounds that raise G16P levels by increasing its production and/or slowing its degradation show promise as candidates for treatment of PMM2-CDG (*Iyer et al., 2019*; *Monticelli et al., 2019*). Whereas higher eukaryotes and some bacteria have a dedicated G16P synthase, in yeast G16P is produced as a dissociated intermediate of the phosphoglucomutase reaction (*Maliekal et al., 2007*; *Neumann et al., 2021*). We hypothesized that evolved mutations in *PGM1* may increase the rate of G16P dissociation. Recent work has shown that a mutation affecting $Mg^{2+}$ coordination in *L. lactis* β-phosphoglucomutase can increase dissociation of β-G16P in vitro (*Wood et al., 2021*). While we did not directly measure G16P synthase activity of Pgm1, we find that intracellular levels of G16P are increased in strains with *pgm1*-T521K or *pgm1*-D295N. We cannot, however, determine if the observed increase in G16P is sufficient to rescue glycosylation. There may be other metabolic changes caused by the *pgm1* mutations that impact Sec53 activity and glycosylation.

A third possible mechanism of compensation is the indirect increase of G16P by increasing the relative flux of the reverse (G6P to G1P) reaction relative to the forward (G1P to G6P) reaction. This would create a futile cycle of G1P/G6P interconversion, which is expected to result in higher intracellular G16P. This hypothesis is attractive in that it accounts for three observations regarding the evolved *pgm1* mutations: (1) the observation of only missense (not nonsense or frameshift) mutations in the evolution experiment, (2) the dominance of suppressors in our genetic assay, and (3) greater extent of suppression for the stability mutant (V238M) of *SEC53*. Nevertheless, the observation that complete loss of Pgm1 enzymatic activity improves growth of *sec53* mutants suggests that there are multiple mechanisms of suppression through *PGM1*.

There is truth to the saying that 'a selection is worth a thousand screens'. Experimental evolution takes what would otherwise be a screen—looking for larger colonies on a field of smaller ones—and turns it into a selection. Here, we demonstrate the power of yeast experimental evolution to identify genetic mechanisms that compensate for the molecular defects of PMM2-CDG disease-associated alleles. This general approach is applicable to other human disease alleles that affect core and conserved biological processes including protein glycosylation, ribosome maturation, mitochondrial function, and RNA modification. Mutations that impinge upon these processes are often highly pleiotropic. In humans, salient phenotypes may be neurological or developmental, but in yeast, these disease alleles invariably lead to slow growth. With a population size of $10^5$, evolution can act on differences as small as 0.001%, far below the ability to resolve on plates. Even our best fluorescence-based assays can only resolve differences of ~0.1% (*Lang and Botstein, 2011*). With a genome size of $10^7$ bp, a mutation rate of $10^{-10}$ per bp per generation, and with continuous selective pressure exerted over thousands of generations, each population will sample hundreds of thousands of coding-sequence mutations. Over longer time scales, experimental evolution can be used to identify rare suppressor mutations, suppressor mutations with modest fitness effects, and/or complex compensatory interactions involving multiple mutations, which are largely absent from current genetic interaction networks.

# Methods

**Key resources table**

| Reagent type (species) or resource | Designation | Source or reference | Identifiers | Additional information |
| --- | --- | --- | --- | --- |
| Recombinant DNA reagent | pTC416-SEC53 | *Lao et al., 2019* | Wild-type SEC53 plasmid (CEN-ARS) | Plasmid map: https://bit.ly/3nBYllD |

*Continued on next page*

*Continued*

| Reagent type (species) or resource | Designation | Source or reference | Identifiers | Additional information |
|---|---|---|---|---|
| Recombinant DNA reagent | pML104-NatMx3 | Addgene (83477) | Cas9 plasmid | Modified plasmids available upon request |
| Recombinant DNA reagent | pAG25 | Addgene (35121) | Integrating plasmid | Modified plasmids available upon request |
| Recombinant DNA reagent | pGEX-2T | Cytiva Life Sciences (28954653) | Bacterial expression plasmid | Modified plasmids available upon request |
| Antibody | anti-carboxypeptidase rabbit polyclonal antibody | Abcam (ab181691) | | (1:1000) |
| Antibody | anti-rabbit IgG HRP-conjugated mouse monoclonal antibody | Cell Signaling Technologies (5127) | | (1:2000) |
| Antibody | anti-invertase goat polyclonal antibody | Novus Biologicals (NB120-20597) | | (1:1000) |
| Antibody | anti-goat IgG HRP-conjugated mouse monoclonal antibody | Rockland Immunochemicals (18-8814-33) | | (1:2000) |

## Experimental evolution

The haploid strains used in the evolution experiment are S288c derivatives and were described previously (*Lao et al., 2019*). Each strain maintains a *URA3*-marked plasmid encoding wild-type *SEC53* (pTC416-SEC53 described in *Lao et al., 2019*). Strains with the p*SEC53*-sec*53*-F126L allele grew too slowly to keep up with a 1:$2^{10}$ dilution every 24 hr.

To initiate the evolution experiment, strains were struck to synthetic defined media (complete supplement mixture [CSM] minus arginine, 0.67% yeast nitrogen base, 2% dextrose) containing 5-Fluoroorotic Acid (5FOA) (Zymo Research, Irvine, CA, USA) to isolate cells that spontaneously lost pTC416-SEC53. A single colony was chosen for each strain and used to inoculate 10 ml of YPD (1% yeast extract, 2% peptone, 2% dextrose) plus ampicillin and tetracycline. These cultures grew to saturation then were distributed into 96-well plates. Every 24-hr populations were diluted 1:$2^{10}$ using a BiomekFX liquid handler into 125 µl of YPD plus 100 µg/ml ampicillin and 25 µg/ml tetracycline to prevent bacterial contamination. Plates were then left at 30°C without shaking. The dilution scheme equates to 10 generations of growth per day at an effective population size of ~$10^5$. Every 100 generations, 50 µl of 60% glycerol was added to each population and archived at −80°C.

$OD_{600}$ was measured every 50 generations over the course of the evolution experiment. After daily dilutions were completed, populations were resuspended by orbital vortexing at 1050 rpm for 15 s then transferred to a Tecan Infinite M200 Pro plate reader for sampling.

## Whole-genome sequencing

Benomyl assays were performed by spotting 5 µl of populations onto YPD plus 20 µg/ml benomyl (dissolved in dimethyl sulfoxide [DMSO]) and incubating at 25°C. Ploidy was then inferred by visually comparing growth of evolved populations to control haploid and diploid strains.

Single clones were isolated for sequencing by streaking 5 µl of cryo-archived populations to YPD agar. One clone from each population was randomly chosen and grown to saturation in 5 ml YPD, pelleted, and frozen at −20°C. Genomic DNA was harvested from frozen cell pellets using phenol–chloroform extraction and precipitated in ethanol. Total genomic DNA was used in a Nextera library preparation as described previously (*Buskirk et al., 2017*). Pooled clones were sequenced using an Illumina NovaSeq 6000 sequencer by the Sequencing Core Facility at the Lewis-Sigler Institute for Integrative Genomics at Princeton University.

## Sequencing analysis

Raw sequencing data were concatenated and then demultiplexed using a dual-index barcode splitter (https://bitbucket.org/princeton_genomics/barcode_splitter/src/master/). Adapter sequences were trimmed using Trimmomatic (*Bolger et al., 2014*). Modified S288c reference genomes were constructed using *ref*orm (https://github.com/gencorefacility/reform; *Khalfan, 2021*) to correct for

strain-construction-related differences between our strains and canonical S288c. Trimmed reads were aligned to these modified S288c reference genomes using BWA and mutations were called using FreeBayes (*Garrison and Marth, 2012*; *Li and Durbin, 2009*). VCF files were annotated with SnpEff (*Cingolani et al., 2012*). All calls were confirmed manually by viewing BAM files in IGV (*Thorvaldsdóttir et al., 2013*). Clones were predicted to be diploids if two or more mutations were called at an allele frequency of 0.5. CNVs and aneuploidies were called via changes in sequencing coverage using Control-FREEC (*Boeva et al., 2012*). CNVs and aneuploidies were confirmed by visually inspecting coverage plots.

## Yeast strain construction

Evolved mutations were reconstructed via CRISPR/Cas9 allele swaps. For mutations that fall within a Cas9 gRNA site (*alg9*-S230R, *pgm1*-T231A, and *pgm1*-G514C), repair templates were produced by PCR-amplifying 500–2000 bp fragments centered around the mutation of interest from evolved clones. Commercially synthesized dsDNA fragments (Integrated DNA Technologies, Coralville, IA, USA) were used as repair templates for all other mutations (*pgm1*-R64C, *pgm1*-S120A, *pgm1*-D295N, *pgm1*-T521K, and *pgm1*Δ). These 500 bp fragments contained our mutation of interest as well as one to two synonymous mutations in the Cas9 gRNA site to halt cutting. The *pgm1*Δ repair template was a dsDNA fragment consisting of 250 bp upstream then 250 bp downstream of the ORF. Repair templates were cotransformed into haploid ancestral strains with a plasmid encoding Cas9 and gRNAs targeting near the mutation site (Addgene #83476) (*Laughery et al., 2015*).

Heterozygous mutant strains were constructed by mating haploid mutants to *MAT*α versions of their respective ancestral strains. Homozygous mutant strains were constructed by reconstructing mutations in *MAT*α versions of the ancestral strains and mating them to the respective *MAT***a** mutant strain. Successful genetic reconstructions were confirmed via Sanger sequencing (Psomagen, Rockville, MD, USA).

Linked *PGM1* strains used for tetrad dissections were constructed by transforming strains with a linearized integrating plasmid (Addgene #35121) containing wild-type *PGM1* (*Goldstein and McCusker, 1999*). Successful integration of the plasmid next to the endogenous *PGM1* locus was confirmed via PCR. Strain information in *Supplementary file 2*.

## Competitive fitness assays

We measured the fitness effect of evolved mutations using competitive fitness assays described previously with some modifications (*Buskirk et al., 2017*). Query strains were mixed 1:1 with a fluorescently labeled version of the p*ACT1-SEC53*-WT ancestral diploid strain. Each p*ACT1-sec53*-F126L strain was outcompeted immediately by the reference at this ratio, so we also tried mixing at a ratio of 50:1. Cocultures were propagated in 96-well plates in the same conditions in which they evolved for up to 50 generations. Saturated cultures were sampled for flow cytometry every 10 generations. Each genotype assayed was done so with at least 24 technical replicates (competitions) of 4 biological replicates (2 clones isolated each from 2 isogenic but separately constructed strains). Analyses were performed in Flowjo and R.

## LOF/non-LOF Bayesian analysis

From reconstruction data from previously published evolution experiments (*Fisher et al., 2018*; *Lang et al., 2013*; *Marad et al., 2018*), we identified ten genes where selection is acting on LOF (*ACE2*, *CTS1*, *ROT2*, *YUR1*, *STE11*, *STE12*, *STE4*$_{haploids}$, *STE5*, *IRA1*, and *IRA2*) and six genes where selection is for non-LOF (*CNE1*, *GAS1*, *KEG1*, *KRE5*, *KRE6*, and *STE4*$_{autodiploids}$). These data established prior probabilities that selection is acting on LOF (0.625) or non-LOF (0.375). We then determined the conditional probabilities of missense and frameshift/nonsense mutations given selection for LOF and non-LOF using 240 mutations across these 16 genes as well as 414 5FOA-resistant mutations at *URA3* and 454 canavanine-resistant mutations at *CAN1* (*Lang and Murray, 2008*). From these data, we can estimate the log likelihood and posterior probabilities that selection is acting on LOF or non-LOF given the observed mutational spectrum for any given gene (*Supplementary file 3*).

## CPY and invertase blots

Strains were grown in 5 ml YPD plus 100 µg/ml ampicillin and 25 µg/ml tetracycline until saturated, then pelleted, and frozen at −20°C. Cell pellets were lysed in Ripa lysis buffer (150 mM NaCl, 1% Nonidet P-40, 0.5% sodium deoxycholate, 0.1% sodium dodecyl sulfate, 25 mM Tris [pH 7.4]) with protease inhibitors (Thermo Fisher Scientific, Waltham, MA, USA) and 10 mM dithiothreitol (DTT), incubated for 30 min on ice, sonicated five times briefly, incubated for a further 10 min on ice, and centrifuged at 20,000 × *g* for 10 min at 4°C. Supernatant was quantified by Micro BCA Protein Assay (Thermo Fisher). 30 µg of each lysate were prepared in Lamelli buffer and were incubated at 95°C for 5 min and chilled at 4°C. Lysates were separated on a 6% or 8% sodium dodecyl sulfate–polyacrylamide gel electrophoresis (SDS–PAGE) gel. Protein was transferred to 0.2 µm pore nitrocellulose at 100 V for 100 min at 4°C in Towbin buffer. The membrane was rinsed and stained with Ponceau S for normalization. The membranes were blocked with 5% milk/TBST for 1 hr at room temperature.

CPY blots were incubated with an anti-carboxypeptidase antibody at 1:1000 overnight at 4°C and washed three times with TBST. The blots were then incubated with anti-rabbit IgG HRP diluted 1:2000 in milk/TBST for 1 hr, washed three times and developed with ECL reagent (Bio-Rad Laboratories, Hercules, CA, USA). Invertase blots were prepared similarly with an anti-invertase antibody and anti-goat IgG HRP. All images were captured on the Bio-Rad ChemiDoc MP Imaging System (Bio-Rad). Analysis was done with Image Lab Software (Bio-Rad). Raw images of blots and Ponceau stain controls are available in *Figure 3—source data 1*.

## Tetrad dissection and inference of genetic interactions

Heterozygous yeast strains (i.e., *SEC53*/*sec53 PGM1*/*pgm1*) were sporulated by resuspending 1 ml of overnight YPD culture in 2 ml of SPO++ (1.5% potassium acetate, 0.25% yeast extract, 0.25% dextrose, 0.002% histidine, 0.002% tryptophan) and incubated on a roller-drum for ~7 days at room temperature. Ten tetrads per strain were dissected on YPD agar and then incubated for 48 hr at 30°C. We performed a log-likelihood test to determine if patterns of segregation are indicative of suppression (*Figure 4—source data 1*). Note that in order to calculate conditional probabilities we had to include an error term (0.01) to account for biological noise in the real data. Our inference of suppression, however, is robust to our choice of error value.

## Bacterial growth and lysis

Wild-type and mutant *PGM1* alleles were cloned into the expression vector pGEX-2T. Cloned vectors were transformed into *E. coli* BL21(DE3). Bacteria were grown in lysogeny broth (LB) medium plus 0.1 mg/ml ampicillin. Each *PGM1* allele was expressed by induction at 0.1 OD with 0.1 mg/ml IPTG for 16 hr at 15°C. Bacteria were harvested by centrifugation, washed with phosphate-buffered saline (PBS), and stored at −80°C. Bacterial lysis was accomplished in 25 mM Tris buffer (pH 8) containing 1 mM EDTA, 2 mM DTT, 5% glycerol, and 0.1 mM phenylmethylsulfonyl fluoride (PMSF), by adding 1 mg/ml lysozyme and 2.5 µg/ml DNase. Clear extract was then obtained by centrifugation.

## Protein purification

Protein purification was accomplished using Glutathione Sepharose High Performance resin (GSTrap by Cytiva, Marlborough, MA, USA) and Benzamidine Sepharose 4 Fast Flow resin (HiTrap Benzamidine FF by Cytiva). Cleavage of the GST-tag was conducted on-column by adding thrombin. All the procedures were performed according to manufacturer's instructions. Briefly, clear extract obtained from 50 ml of bacterial culture was loaded onto the GSTrap column (5 ml), previously equilibrated with 25 mM Tris (pH 8) containing 2 mM DTT and 5% glycerol. Unbound protein was eluted, then the column was washed with 20 mM Tris (pH 8) containing 1 mM DTT, 150 mM NaCl, 1 mM MgCl$_2$, and 5% glycerol (buffer A). Thrombin (30 units in 5 ml of buffer A) was loaded and the column was incubated for 14–16 hr at 10°C. GSTrap and HiTrap Benzamidine columns were connected in series, and elution of the untagged protein was accomplished with buffer A. Fractions were analyzed by SDS–PAGE and activity assays. Active fractions were finally collected and concentrated by ultrafiltration. A yield spanning 1.0–4.6 mg of protein per liter of bacterial culture was obtained.

Each protein was obtained in a pure, monomeric form as judged by SDS–PAGE and gel filtration analyses, with a single band observed at the expected molecular weight (63 kDa). Gel filtration

analysis was performed on BioSep-SEC-S 3000 column (Phenomenex, Torrance, CA, USA) at 1 ml/min. The eluent was 20 mM Tris (pH 8), 1 mM MgCl$_2$, and 150 mM NaCl.

## Activity assays

Phosphoglucomutase activity was assayed spectrophotometrically at 340 nm and 29°C by following the reduction of NADP$^+$ to NADPH in a 0.3 ml reaction mixture containing 25 mM Tris (pH 8), 5 mM MgCl$_2$, 1 mM DTT, 0.25 mM NADP$^+$, and 2.7 U/ml glucose-6-phosphate dehydrogenase in the presence of 0.3 mM G1P and 0.02 mM G16P. Additional experiments were performed with the addition of M1P (10, 50, and 200 μM). $K_m$ were evaluated by measuring activity in the presence of G1P ranging from 0 mM to 0.6 mM. Analyses were performed in R using the *drc* package (*Ritz et al., 2015*). Bisphosphatase activity was measured spectrophotometrically at 340 nm and 29°C in the presence of 0.1 mM G16P, 0.25 mM NADP$^+$, and 2.7 U/ml glucose-6-phosphate dehydrogenase. Activity expressed as the number of micromoles of substrate transformed per minute per mg of protein under the standard conditions.

Thermostabilities of Pgm1 were determined by incubating purified protein for 5 or 10 min at different temperatures (30, 35, 40, and 45°C) in 20 mM Tris (pH 8), 1 mM MgCl$_2$, 150 mM NaCl, 5% glycerol, and 0.1 mg/ml BSA. After cooling on ice, residual activity was measured under standard conditions.

## $^{31}$P-NMR spectroscopy

Phosphoglucomutase and phosphomannomutase activity were measured by $^{31}$P-NMR spectroscopy (*Citro et al., 2017*). This assay allows us to exclude the effects on the change in absorbance due to impurities that could be substrates for the ancillary enzymes required for the assay. This is of utmost importance when the activity is particularly low and measuring the rate of change in absorbance requires long incubation times, as it is the case of phosphomannomutase activity for Pgm1. A discontinuous assay was performed. Appropriate amounts of proteins were incubated at 30°C with 1 mM G1P or M1P in 20 mM Tris (pH 8), 1 mM MgCl$_2$, 0.1 mg/ml BSA, in the presence of 20 μM G16P for up to 60 min. The reaction was stopped with 50 mM EDTA on ice and heat inactivation. The content of the residual substrate (M1P or G1P) and/or the product (M6P or G6P) were measured recording $^{31}$P-NMR spectra. Creatine phosphate was added as an internal standard for the quantitative analysis. The $^1$H-decoupled, one-dimensional $^{31}$P spectra were recorded at 161.976 MHz on a Bruker Avance III HD spectrometer 400 MHz, equipped with a BBO BB-H&F-D CryoProbe Prodigy fitted with a gradient along the *z*-axis, at a probe temperature of 27°C. Spectral width 120 ppm, delay time 1.2 s, and pulse width of 12.0 μs were applied. All the samples contained 10% $^2$H$_2$O for internal lock.

Similarly, forward (G1P to G6P) and reverse (G6P to G1P) phosphoglucomutase activities were measured following the same procedure described above, with 1 mM G1P or G6P as the substrate.

## Quantification of G16P abundance

Yeast metabolite extraction was performed as described previously (*Gonzalez et al., 1997*). Briefly, five replicate cultures per genotype were grown in 5 ml YPD plus 100 μg/ml ampicillin and 25 μg/ml tetracycline at 30°C. 10$^7$ cells of mid-log culture were collected on nylon filters, washed with 10 ml cold PBS, frozen in liquid nitrogen, and lyophilized. Cells were then treated with an ice-cold solution containing 14% HClO$_4$ and 91 mM imidazole, frozen and thawed five times (dry ice/acetone followed by 40°C bath) with vigorous shaking between each cycle, then centrifuged at 13,500 × *g* at 4°C. Supernatant was neutralized with 3 M K$_2$CO$_3$ and centrifuged again at 13,500 × *g* at 4°C to eliminate salt precipitate.

G16P in the metabolite extracts was measured through stimulation of rabbit muscle phosphoglucomutase as described previously (*Veiga-da-Cunha et al., 2008*). Phosphoglucomutase activity was assayed spectrophotometrically at 340 nm, in a mixture containing 87 mM Tris (pH 7.6), 5.6 mM MgCl$_2$, 0.11 mM ethylene glycol tetraacetic acid (EGTA), 0.1 mg/ml BSA, 0.048 mM NADP$^+$, 0.48 mM G1P, 0.26 U/ml glucose-6-phosphate dehydrogenase, at 36°C. G1P had previously been purified from G16P through anionic exchange chromatography on AG1x8 Hydroxide Form column in step gradient of triethylammonium bicarbonate from 0.01 M to 1 M (*Monticelli et al., 2019*). Three technical replicates for each of the five biological replicates were assayed. 0.0–0.15 μM commercial G16P (Merck & Co, Rahway, NJ, USA) was used as the standard.

## Acknowledgements

We thank members of the Andreotti Lab and members of the Lang Lab for comments on the manuscript. We thank M Kathryn Iovine for sharing the pGEX-2T plasmid. We thank Lesa Beamer for useful discussions. This study was supported by the National Institutes of Health: R01GM127420 to GIL and P20GM139769 to RS (Trudy Mackay PI/PD). The content is solely the responsibility of the authors and does not necessarily represent the official views of the National Institutes of Health. This study was also supported by DiSTABiF, University of Campania 'Luigi Vanvitelli' (fellowship POR Campania FSE 2014/2020 'Dottorati di Ricerca Con Caratterizzazione Industriale' to MA) and the Short Term Mobility Program 2022, CNR to GA. Portions of this research were conducted on Lehigh University's Research Computing infrastructure partially supported by the National Science Foundation (Award 2019035). Perlara PBC provided funding support for reagents and personnel time.

## Additional information

### Competing interests

Ethan O Perlstein: The other authors declare that no competing interests exist.

### Funding

| Funder | Grant reference number | Author |
|---|---|---|
| National Institutes of Health | R01GM127420 | Gregory I Lang |
| National Institutes of Health | P20GM139769 | Richard Steet |

The funders had no role in study design, data collection, and interpretation, or the decision to submit the work for publication.

### Author contributions

Ryan C Vignogna, Conceptualization, Formal analysis, Investigation, Visualization, Writing – original draft, Writing – review and editing; Mariateresa Allocca, Maria Monticelli, Formal analysis, Investigation, Writing – review and editing; Joy W Norris, Investigation; Richard Steet, Supervision, Writing – review and editing; Ethan O Perlstein, Conceptualization, Funding acquisition, Writing – review and editing; Giuseppina Andreotti, Conceptualization, Formal analysis, Supervision, Writing – review and editing; Gregory I Lang, Conceptualization, Formal analysis, Supervision, Funding acquisition, Visualization, Writing – original draft, Writing – review and editing

### Author ORCIDs

Ryan C Vignogna http://orcid.org/0000-0001-5943-6464
Mariateresa Allocca http://orcid.org/0000-0003-3693-2515
Maria Monticelli http://orcid.org/0000-0003-3136-2138
Richard Steet http://orcid.org/0000-0002-0975-4963
Ethan O Perlstein http://orcid.org/0000-0002-4734-4391
Giuseppina Andreotti http://orcid.org/0000-0002-1594-0156
Gregory I Lang http://orcid.org/0000-0002-7931-0428

### Decision letter and Author response

Decision letter https://doi.org/10.7554/eLife.79346.sa1
Author response https://doi.org/10.7554/eLife.79346.sa2

## Additional files

### Supplementary files
- Supplementary file 1. List of mutations that arose in the evolution experiment.
- Supplementary file 2. Genotypes of strains used in this study.

- Supplementary file 3. Loss-of-function (LOF)/non-LOF Bayesian analysis.
- MDAR checklist

### Data availability

The short-read sequencing data reported in this study have been deposited to the NCBI BioProject database, accession number PRJNA784975.

The following dataset was generated:

| Author(s) | Year | Dataset title | Dataset URL | Database and Identifier |
|---|---|---|---|---|
| Vignogna RC, Lang GI | 2022 | Experimentally-evolved *Saccharomyces cerevisiae* clones | https://www.ncbi.nlm.nih.gov/bioproject/?term=PRJNA784975 | NCBI BioProject, PRJNA784975 |

The following previously published datasets were used:

| Author(s) | Year | Dataset title | Dataset URL | Database and Identifier |
|---|---|---|---|---|
| Lang, et al | 2013 | The sequencing of *Saccharomyces cerevisiae* strains | https://www.ncbi.nlm.nih.gov/bioproject/?term=PRJNA205542 | NCBI BioProject, PRJNA205542 |
| Fisher, et al | 2018 | Evolved Autodiploid Clones | https://www.ncbi.nlm.nih.gov/bioproject/?term=PRJNA422100 | NCBI BioProject, PRJNA422100 |
| Johnson, et al | 2021 | Evolved *S. cerevisiae* population sequencing | https://www.ncbi.nlm.nih.gov/bioproject/?term=PRJNA668346 | NCBI BioProject, PRJNA668346 |

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
