## [Editor Report]

This valuable paper shows that experimental evolution can shed new and unbiased light on mutations involved in human diseases by showing how growth defects can be compensated. The evidence is convincing, benefiting from not only genetics but also well-established biochemical assays. This paper will be of interest to a broad group of evolutionary biologists and biologists interested in human diseases.

---

## [Decision Letter]

**Decision letter after peer review:**

Thank you for submitting your article "Experimental evolution of phosphomannomutase-deficient yeast reveals compensatory mutations in a phosphoglucomutase" for consideration by *eLife*. Your article has been reviewed by 3 peer reviewers, including Wenying Shou as Reviewing Editor and Reviewer #3, and the evaluation has been overseen by George Perry as the Senior Editor. The following individual involved in review of your submission has agreed to reveal their identity: Hudson H. Freeze (Reviewer #2);

Essential revisions:

Please determine whether Sec53 and Pgm1 proteins directly interact in yeast and whether the mutations to study are on the interaction interface.

That the single-substitutions did not capture the fitness gains as much as the primary strains suggests genetic connections among the suppressors, and you should test the combinatorial impact of mutations. Over 90% of the yeast binary genetic interactions have been discovered. A simple computational screen using tools available at SGD could explore the possibility of rewiring the genetic interactions as a mode of action. Some of these combinations may be tested as well, but is not absolutely required.

EOP's observation that an aldose reductase inhibitor (epalrestat) has remarkable effects on a single patient and that the drug is now entering clinical trials is a real breakthrough. One of the observations is that Glc1,6P, a stabilizer of PMM2, is increased in PMM2 patient cells treated with the drug. Presumably, any way to do this would have a benefit. You discuss this, but the critical experiment is missing: measuring Glc1,6 P in the cells you study. Regardless of the results, it will be informative to perform this measurement, to learn whether pgm1 variants increase the steady state level of Glc1,6-bisphosphate. If they do, the level in itr2 mutants should be measured as well.

Figure 3 CPY gel should be rerun to prevent bands from nearly running off the gel.

*Reviewer #1 (Recommendations for the authors):*

1. Based on the data obtained between pACT1 and pSEC53-driven expression of the SEC53 mutant alleles, the pattern of suppressors appears to be different. Authors report that the variants expressed from strong pACT1 promoters show more suppressors than those driven by native promoters. Is this a general trend in experimental evolution that slower-growing strains tend to show lesser suppressors?

2. It isn't clear whether the strains used for evolution experiments harbor genomically encoded copies of variants or plasmid-borne copies (CEN or 2 micron)?

3. Authors use the term "nearly-identical homolog" twice in the article. The databases such as InParanoid show a clear orthology relationship. What does nearly-identical mean? Are the authors referring to sequence identity?

4. Page 8, line 27 and in the following paragraphs, the authors describe that pgm1 mutations show a dominant phenotype, including pgm1Δ. However, in the latter part of the manuscript, the pgm1Δ genotype is described as not dominant. Also, the authors haven't described if Pgm1 protein is a monomer or a dimer? These scenarios may explain the lack of dominant phenotype observed in the case of pgm1Δ.

5. Page 3, line 67. Reference to previous work is enough. Consider rewriting the sentence to omit " one of us (E.O. Perlstein)."

*Reviewer #2 (Recommendations for the authors):*

This is an impressive and innovative study that may have relevance to future treatments of other mutations in PMM2-CDG patients. The reduced activity of PGM1 was only seen in mutation that increases enzyme stability of a homodimer. Most patients are compound heterozygotes. So, it is difficult to tell what the therapeutic benefits might be. There is no definitive mechanism, but the authors suggest a number of reasonable approaches to identify one. Based on the impressive results of epalrestat treatment in PMM2-CDG cells and patients, the study could be strengthened by showing that the PGM1 variants increase the steady state amount/concentration of Glc1,6 bisphosphate. If that were true, other PMM2 variants could be tested directly. While the authors did investigate the activity of PGM1 for generating or decreasing Glc1,6 bisphosphate using purified enzyme in vitro, that is not the same as determining the steady state levels.

Other points:

What is the glucose concentration of rich glucose media? Is that concentration important? Are the results different if glucose is reduced?

The ALG9 and ALG12 mutations are puzzling. Do they decrease the size of the dolichol polysaccharide precursor as they would in ALG9- and ALG12-CDG patients?

In figure 2, ITR2 is a significant candidate. Since this is also a "monosaccharide" transporter, could it possibly alter polyol metabolism or flux? Could it possibly alter the amount of Glc1,6 bisphosphate? If that co-factor increases in the PGM1 variants, that should also be checked in the ITR2 clones.

In Figure 3, the CPY gel should be rerun so that the bands are not nearly off the gel. Is CPY nearly gone in the +/+ lane or has it run off the gel? Hard to tell.

In Figure 5, s4, what does "known periods of time" mean?

Figure 5, s5…How long was that incubation?

[Editors’ note: further revisions were suggested prior to acceptance, as described below.]

Thank you for resubmitting your work entitled "Evolutionary rescue of phosphomannomutase deficiency in yeast models of human disease" for further consideration by *eLife*. Your revised article has been evaluated by George Perry (Senior Editor) and a Reviewing Editor.

The manuscript has been improved but there are some remaining issues that need to be addressed, as outlined below:

*Reviewer #3 (Recommendations for the authors):*

The authors responded to my question about the steady state levels of Glc1,6,P by measuring it in a coupled assay where its amount was the limiting component. The result shows a statistical increase of this molecule in several of the strains. However, it is difficult to determine whether this increase is sufficient to account for the rescue of the glycosylation phenotype.

*Reviewer #4 (Recommendations for the authors):*

When I wrote "pre-existing mutations", I meant mutations that already existed when you grew up the culture to do the evolution experiment. The question can be changed to: how much of the phenotypic changes are from very early stage mutations? Do you need 1000 generations?

---

## [Author Response]

Essential revisions:Please determine whether Sec53 and Pgm1 proteins directly interact in yeast and whether the mutations to study are on the interaction interface.

Systematic studies of protein-protein interactions have been carried out in yeast using various methods (doi.org/10.1093/nar/gky1079) and no physical interaction between Pgm1 and Sec53 has been identified. We also performed computational protein-protein interaction analyses using AlphaFold-multimer (doi.org/10.1101/2021.10.04.463034) and no interaction between Pgm1 and Sec53 was predicted. Additionally, we think it is unlikely any physical interaction would involve the mutated residues, given that the evolved *pgm1* mutations occur around the active site of the enzyme.

That the single-substitutions did not capture the fitness gains as much as the primary strains suggests genetic connections among the suppressors, and you should test the combinatorial impact of mutations. Over 90% of the yeast binary genetic interactions have been discovered. A simple computational screen using tools available at SGD could explore the possibility of rewiring the genetic interactions as a mode of action. Some of these combinations may be tested as well, but is not absolutely required.

No mutation that cooccurs with any *pgm1* mutation fall within genes known to genetically interact with *PGM1*, save one mutation in *FRA1*. It is likely that evolved mutations do not reciprocate the effects of gene deletions and the fitness gains we observe are due to allele-specific genetic interactions. We have added this to the Discussion:

“…and only one evolved *pgm1* clone contains a mutation in a known *PGM1* interactor (*FRA1*). These suggest any putative genetic interactions with our evolved *pgm1* mutations are allele-specific.”

EOP's observation that an aldose reductase inhibitor (epalrestat) has remarkable effects on a single patient and that the drug is now entering clinical trials is a real breakthrough. One of the observations is that Glc1,6P, a stabilizer of PMM2, is increased in PMM2 patient cells treated with the drug. Presumably, any way to do this would have a benefit. You discuss this, but the critical experiment is missing: measuring Glc1,6 P in the cells you study. Regardless of the results, it will be informative to perform this measurement, to learn whether pgm1 variants increase the steady state level of Glc1,6-bisphosphate. If they do, the level in itr2 mutants should be measured as well.

We performed the suggested experiment, measuring glucose-1,6-bisphosphate levels in several of our reconstructed strains. We find increased G16P concentration in strains carrying one of two *pgm1* mutations we tested (T521K and D295N). These findings have been added to the Results:

“We next assessed G16P levels for two *pgm1* mutant alleles in both the p*ACT1*-*sec53*-V238M and the p*ACT1*-*SEC53*-WT backgrounds (Figure 5 — figure supplement 5). We find that strains heterozygous for *pgm1*-T521K or *pgm1*-D295N show statistically significant increases in the amount of G16P present compared to wild-type *PGM1* in the *sec53*-V238M background (18.4 and 21.1 pmol/million cells, respectively, *W*=24 and 8, p=0.003 and 0.0002, respectively, Mann–Whitney U test). However, this difference is not significant in the *SEC53*-WT background (*W*=41 and 15, p=0.35 and 0.078, respectively, Mann–Whitney).”

and Discussion:

“While we did not directly measure G16P synthase activity of Pgm1, we find that intracellular levels of G16P are increased in strains with *pgm1*-T521K or *pgm1*-D295N, consistent with the hypothesis of increased dissociation of the G16P intermediate.”

We have also added a new figure supplement corresponding to these data (Figure 5 — figure supplement 4) and a Methods section.

Figure 3 CPY gel should be rerun to prevent bands from nearly running off the gel.

We have rerun this gel and updated Figure 3 and the corresponding source data.

Reviewer #1 (Recommendations for the authors):1. Based on the data obtained between pACT1 and pSEC53-driven expression of the SEC53 mutant alleles, the pattern of suppressors appears to be different. Authors report that the variants expressed from strong pACT1 promoters show more suppressors than those driven by native promoters. Is this a general trend in experimental evolution that slower-growing strains tend to show lesser suppressors?

The patterns of suppression differed between *sec53*-F126L and *sec53*-V238M clones, but we did not sequence populations evolved with the endogenous promoter.

2. It isn't clear whether the strains used for evolution experiments harbor genomically encoded copies of variants or plasmid-borne copies (CEN or 2 micron)?

Every variant used in this study (*sec53, pgm1,* or otherwise) is integrated into the genome. We clarified this by including a strain genotype table (Supplementary File 2)

3. Authors use the term "nearly-identical homolog" twice in the article. The databases such as InParanoid show a clear orthology relationship. What does nearly-identical mean? Are the authors referring to sequence identity?

We agree that this terminology is vague. We now simply refer to them as “homolog”.

4. Page 8, line 27 and in the following paragraphs, the authors describe that pgm1 mutations show a dominant phenotype, including pgm1Δ. However, in the latter part of the manuscript, the pgm1Δ genotype is described as not dominant.

We performed two phenotypic assays: fitness assays and tetrad analysis. The *pgm1*Δ is dominant in the fitness assays but recessive in the tetrad analyses. In contrast, the *pgm1* suppressor mutations are dominant in both assays.

Also, the authors haven't described if Pgm1 protein is a monomer or a dimer? These scenarios may explain the lack of dominant phenotype observed in the case of pgm1Δ.

Pgm1 is a known monomer and we verified this by SDS-PAGE analysis. We have added this information to the Methods:

“Each protein was obtained in a pure, monomeric form as judged by SDS-PAGE and gel filtration analyses, with a single band observed at the expected molecular weight (63 kDa).”

5. Page 3, line 67. Reference to previous work is enough. Consider rewriting the sentence to omit " one of us (E.O. Perlstein)."

We made the suggested edited.

Reviewer #2 (Recommendations for the authors):This is an impressive and innovative study that may have relevance to future treatments of other mutations in PMM2-CDG patients. The reduced activity of PGM1 was only seen in mutation that increases enzyme stability of a homodimer. Most patients are compound heterozygotes. So, it is difficult to tell what the therapeutic benefits might be. There is no definitive mechanism, but the authors suggest a number of reasonable approaches to identify one. Based on the impressive results of epalrestat treatment in PMM2-CDG cells and patients, the study could be strengthened by showing that the PGM1 variants increase the steady state amount/concentration of Glc1,6 bisphosphate. If that were true, other PMM2 variants could be tested directly. While the authors did investigate the activity of PGM1 for generating or decreasing Glc1,6 bisphosphate using purified enzyme in vitro, that is not the same as determining the steady state levels.

We extracted metabolites and quantified glucose-1,6-bisphosphate levels in our reconstructed strains. These data are consistent with the hypothesis that the *pgm1* suppressor mutations increase G16P levels in the *sec53*-V238M background. We have added new sections to the Results, Discussion, and Methods, as discussed above.

Other points:What is the glucose concentration of rich glucose media? Is that concentration important? Are the results different if glucose is reduced?

The glucose concentration is a yeast-microbiology-standard 2%. We have added this information to the Methods. We do not expect the selective pressure due to the glycosylation defect to be significantly affected by the glucose concentration.

The ALG9 and ALG12 mutations are puzzling. Do they decrease the size of the dolichol polysaccharide precursor as they would in ALG9- and ALG12-CDG patients?

In the present study we examined the mechanism for *pgm1* suppression. However, we agree that the *alg9*-S230R and *alg*12*-*Y41H mutations are interesting as mutations in these genes also cause CDG in humans. Whether the mechanism of sec53 suppression is similar to the mechanism underlying pathogenesis is an open question.

In figure 2, ITR2 is a significant candidate. Since this is also a "monosaccharide" transporter, could it possibly alter polyol metabolism or flux? Could it possibly alter the amount of Glc1,6 bisphosphate? If that co-factor increases in the PGM1 variants, that should also be checked in the ITR2 clones.

*ITR2* may in fact be a suppressor mutation; however, the four *ITR2* mutations are identical and, therefore, may have arose in the starting inoculum prior to the evolution experiment. The same is true for several two other mutations (*EXO70* and *MIC10*). We have therefore removed these genes from the heatmap. To answer the second part of the question, the *itr2* mutations do not co-occur with any of the *pgm1* mutations.

In Figure 3, the CPY gel should be rerun so that the bands are not nearly off the gel. Is CPY nearly gone in the +/+ lane or has it run off the gel? Hard to tell.

We reran the gel and have updated Figure 3.

In Figure 5, s4, what does "known periods of time" mean?

Either 0 or 60 minutes. We have added this information to the figure legend (now Figure 5 — figure supplement 3).

Figure 5, s5…How long was that incubation?

Either 0, 5, 10, or 15 minutes. We have added this information to the figure legend.

[Editors’ note: further revisions were suggested prior to acceptance, as described below.]

Reviewer #3 (Recommendations for the authors):The authors responded to my question about the steady state levels of Glc1,6,P by measuring it in a coupled assay where its amount was the limiting component. The result shows a statistical increase of this molecule in several of the strains. However, it is difficult to determine whether this increase is sufficient to account for the rescue of the glycosylation phenotype.

We agree that is difficult to determine the extent to which the observed increase in G16P is sufficient to rescue glycosylation. We cannot rule out that there may be other metabolic changes caused by the *pgm1* mutations that would impact Sec53 (Pmm2) activity and glycosylation. In the Discussion we are careful not to assert causality, rather we describe this result as “consistent with the hypothesis” that evolved Pgm1 mutants increase the dissociation of G16P, thereby stabilizing the pathogenic Sec53 variant.

To make this point clearer, we added to the Discussion:

“We cannot, however, determine if the observed increase in G16P is sufficient to rescue glycosylation. There may be other metabolic changes caused by the pgm1 mutations that impact Sec53 activity and glycosylation.”

Reviewer #4 (Recommendations for the authors):When I wrote "pre-existing mutations", I meant mutations that already existed when you grew up the culture to do the evolution experiment. The question can be changed to: how much of the phenotypic changes are from very early stage mutations? Do you need 1000 generations?

The *pgm1* mutations were likely not in the original population, nor were they likely the first mutations to arise (as evidenced by their appearance in populations following audodiploidization). The dynamics in Figure 1C show that most of the *sec53*-V238M populations underwent a large change in saturation density prior to Generation 500, suggesting that fewer generations may indeed have been sufficient to identify *pgm1* suppressor mutations. However, it is worth noting that *pgm1* is not the only beneficial mutation and some populations increased saturation density much later.

In general, the number of generations necessary to identify suppressor mutations will depend on the mutation rate, the selective benefit of those mutations, and the population size. Under the regime we use, the first selective sweep usually occurs in the first several hundred generations (Lang *et al.* 2013). It is worth noting, however, that rare suppressor mutations, suppressor mutations with modest fitness effects, and/or complex compensatory interactions involving multiple mutations may only be detected over longer experimental time scales.

To clarify the use of longer time-scales we edited the final sentence of the Discussion:

“Over longer time-scales, experimental evolution can be used to identify rare suppressor mutations, suppressor mutations with modest fitness effects, and/or complex compensatory interactions involving multiple mutations, which are largely absent from current genetic interaction networks.”